



# Retrieval pseudo BRDF-adjusted surface reflectance at 440nm from Geostationary Environmental Monitoring Spectrometer (GEMS)

Suyoung Sim[1], Sungwon Choi[2], Daeseong Jung[1], Jongho Woo[1], Nayeon Kim[1], Seongwoo Park[1], Honghee Kim[1], Ukkyo Jeong[1], Hyunkee Hong[3], and Kyung-soo Han[1]

[1]Division of Earth Environmental System Science (Major of Spatial Information System Engineering), Pukyong National University, Busan, Republic of Korea
[2]BK21 FOUR Project of the School of Integrated Science for Sustainable Earth Environmental Disaster, Pukyong National University, Busan, Republic of Korea
[3]Environmental Satellite Center, National Institute of Environmental Research, Incheon, Republic of Korea

**Correspondence:** Kyung-soo Han (kyung-soo.han@pknu.ac.kr)

**Abstract.** In remote sensing applications, enhancing the precision of level 2 (L2) algorithms relies heavily on the accurate estimation of the surface reflectance across the ultraviolet (UV) to visible (VIS) spectrum. However, the mutual dependence between the L2 algorithms and surface reflectance retrieval poses challenges, necessitating an alternative approach. To address this issue, many satellite algorithms generate Lambertian Equivalent Reflectance (LER) products as a priori surface reflectance data; however, this often results in an underestimation of these data. This study introduces a novel approach to surface reflectance retrieval, termed background surface reflectance (BSR), which leverages a semi-empirical Bidirectional Reflectance Distribution Function (BRDF) model to simulate surface reflectance based on BRDF components. This study pioneered the application of the BRDF model to hyperspectral satellite data in the UV-VIS region, aiming to provide more realistic preliminary surface reflectance data. In this styudy, the Geostationary Environment Monitoring Spectrometer (GEMS) data was used, and a comparative analysis of the GEMS BSR and GEMS LER retreieved in this study revealed an improvement in the relative Root Mean Squared Error (rRMSE) accuracy of 3%. Additionally, a time-series analysis across diverse land types indicated a greater stability exhibited by the BSR than by the LER. For further validation, the BSR was compared with other LER databases using ground-truth data, yielding superior simulation performance. These findings present a promising avenue for enhancing the accuracy of surface reflectance retrieval from hyperspectral satellite data, thereby advancing the practical applications of remote sensing algorithms.

## 1 Introduction

Surface reflectance, the fraction of solar radiation reflected by the Earth's surface, is a key parameter in meteorology, environmental studies, and climate research (Dickinson, 1983). Because surface reflectance is utilized in remote- sensing systems to derive various geophysical, chemical, and biological variables (Veefkind et al., 2006; Noguchi et al., 2014), accurate satellite observations of land surface reflectance are essential for developing accurate remote-sensing algorithms. Ultraviolet (UV)-to-visible (VIS) surface reflectance is of great importance for atmospheric component retrieval algorithms and related studies,




especially as an input to various level 2 (L2) algorithms, including aerosols, clouds, ozone, and gas tracers (Lin et al., 2015; Lorente et al., 2018).

However, it is difficult to determine the calculation precedence between these L2 algorithms and surface reflection algorithms. Surface reflectance is the fundamental input data for other L2 algorithms, and L2 data are also essential for surface reflectance

retrieval algorithms. In the calculation of surface reflectance, aerosol optical depth (AOD) and atmospheric gas products (such as ozone, total precipitable water (TPW), nitrogen dioxide ($NO_2$), etc.) are fundamental parameters. Simultaneously, surface reflectance is crucial for the calculation of these atmospheric constituents and forms an essential component of their output. This reciprocal relationship has a significant effect on the accuracy of the calculated data. Therefore, a background field for AOD and atmospheric products should be established when calculating surface reflectance, or conversely, a background field for surface

reflectance should be developed when calculating AOD and atmospheric substances. Therefore, several studies have calculated and applied alternative surface reflectance data to overcome this dilemma.

Most satellite algorithms that focus on observing the UV-VIS region, such as the Total Ozone Mapping Spectrometer (TOMS) (Herman and Celarier, 1997), Global Ozone Monitoring Experiment (GOME) (Koelemeijer et al., 2003), and Ozone Monitoring Instrument (OMI) (Kleipool et al., 2008) produce a priori surface reflectance products called Lambertian equivalent reflectance

(LER). The LER archive is a climatology database calculated using the minimum reflectance method under the assumption of a Lambertian surface. The minimum reflectance technique uses the lowest observed ground reflectance for the same pixel within the compositing period. This technique assumes that the minimum value of the surface reflectance generated during the synthesis period minimizes the effects of the atmosphere and clouds, and adopts it as a stable value in a clear sky. This method has the advantage of being simple to implement, but has a limitation in that it cannot consider realistic surface reflection

properties and can easily underestimate the actual surface reflectance. Occasionally, overcalculations can occur because of a failure to reflect the characteristics of changes in the indicators in real time.

Under- and over-estimated surface reflectance arising from neglecting surface anisotropy can significantly compromise the accuracy of other satellite-derived products. For instance, an underestimated surface reflectance may result in an overestimation of the AOD, affecting the accuracy of aerosol concentration estimates. Conversely, if the surface reflectance is overestimated,

the opposite effects would occur. Based on prior studies, the variation of the surface reflectance 0.05 in the blue channel can lead underestimation of approximately -0.17 in the range where the AOD is less than 0.4 (Li et al., 2012). Additionally, and an error of 0.01 while in the surface reflectance in the UV region results in an AOD error of 0.1 (Veefkind et al., 2000). In addition, errors in cloud retrieval can also occur, affecting cloud properties such as cloud fraction and optical thickness. According to (Lorente et al., 2018), clear-sky air mass factors (AMF) are up to 20% higher and cloud radiance fractions up to 40% lower

if surface anisotropic reflectance is considered. Furthermore, biases may arise in the estimation of trace gas concentrations, affecting the reliability of the $NO_2$ and $SO_2$ measurements, which are sensitive to AMF. According to (Noguchi et al., 2014), neglecting surface reflectance anisotropy can lead to AMF errors of more than 10%, particularly in areas with high $NO_2$ concentrations near the surface. These issues highlight the critical role of accurate surface reflectivity data in ensuring the reliability of satellite-derived atmospheric and surface properties. Therefore, several studies have claimed that anisotropic effects,



such as the bidirectional reflectance distribution function (BRDF), should be considered in satellite-based cloud and trace gas retrieval (Lin et al., 2015; Vasilkov et al., 2017).

Therefore, increasing recognition of the importance of considering surface anisotropy in surface reflectance retrieval has led to various research efforts. These efforts have resulted in institutions producing and providing products that account for the BRDF effects based on existing LER databases. This is exemplified by the geometry-dependent surface Lambertian-equivalent

reflectivity (GLER) and the directionally dependent Lambertian-equivalent reflectivity (DLER). The National Aeronautics and Space Administration (NASA) provides a GLER database reprocessed with the Moderate Resolution Imaging Spectroradiometer (MODIS) BRDF based on the OMI LER (Qin et al., 2019). This dataset provides the reflectance considering the BRDF effects at 440 and 466 nm by applying the MODIS BRDF products to the existing LER database. An advantage of this dataset is its direct applicability to OMI observation scenes, enabling its use in calculating the reflectance for every scene observed by the

OMI. The wavelength range provided by MODIS BRDF used in GLER retrieval is from 459 to 479 nm, which encompasses the 466 nm output from OMI GLER, allowing for direct application of these data. However, the 440 nm wavelength falls outside the range covered by MODIS, making it impossible to use these data directly. To address this, the ratio of LER at 466 and 440 nm was used to apply the MODIS BRDF data to the calculation of GLER. Concerns arise from this method because it may overlook the BRDF dependency on certain wavelengths, assuming linear behavior across wavelengths.

In addition, the Satellite Application Facility on Atmospheric Composition Monitoring (AC SAF) and the European Space Agency (ESA) provided a DLER database that considers the viewing geometry from GOME-2 (Tilstra et al., 2021) and the Tropospheric Monitoring Instrument (TROPOMI) (Tilstra et al., 2023). The DLER database introduced by (Tilstra et al., 2021) considers anisotropic features at various viewing angles as an advantage of polar and sun-synchronous orbit satellites. GOME-2 and TROPOMI DLER calculations were performed using the regression coefficients calculated for the 5 and 9 view containers,

respectively. However, these data are constructed from climatology, such as the LER database, and are not updated annually but are provided only once a month, making it difficult to reflect the changing land surface characteristics in real time. In addition, there are limitations to reflecting the characteristics of indicators that change every time with a single fixed coefficient, and it is difficult to consider the influence of other geometric conditions (such as the solar zenith angle).

In this study, we suggest the retrieval of an alternative pseudo-BRDF-adjusted surface reflectance called background surface

reflectance (BSR). This approach reflects both the high temporal resolution of GLER and the advantages of DLER's own BRDF consideration, while solving the limitations of both. This algorithm concept has now been applied to various satellites to retrieve surface albedo, which must simulate multiple angles (Schaaf et al., 2002; Lee et al., 2018), and cloud detection (Kim et al., 2017; Yeom et al., 2020). However, UV–VIS satellites, particularly hyperspectral satellites, have not yet been investigated . Therefore, in this study, we propose, for the first time, the application of the BRDF model to hyperspectral satellite data by

observing the UV-VIS region for more realistic preliminary surface reflectance data. In this study, we focused on 440 nm, which is used as an input from $NO_2$, clouds, and aerosols. . This algorithm consists of two main steps: atmospheric correction, BRDF modeling and BSR retrieval. A detailed description of each step is provided in Section 3.



## 2 Study area and Materials

### 2.1 Study area

This study focuses on of this study encompasses Northeast Asia, which extends from 20° to 45° north latitude and 100° to 140° east longitude. The temporal scope of the study spans one year, specifically from January 1 to December 31, 2021. This spatial and temporal framework provides a comprehensive basis for analyzing and understanding the atmospheric and environmental dynamics within this region over a specified period.

### 2.2 Geostationary Environment Monitoring Spectrometer (GEMS)

A Geostationary Environment Monitoring Spectrometer (GEMS) is a hyperspectral spectrometer mounted on GeoKompsat2B (GK-2B) that covers the UV-VIS region (300–500 nm) with a full width at half maximum (FWHM) 0.6 nm. It maintains a spatial resolution of 7 × 8 km for gas outputs and 3.5 × 8 km for other outputs per pixel at Seoul (Choi et al., 2018; Kim et al., 2020). The primary objective of the GEMS is to provide air quality (AQ) components, such as ozone, aerosols, and gas tracers, at high temporal and spatial resolutions. In this study, GEMS-derived Level-1C data (radiance and irradiance) (Kang et al.,

2020, 2021) were employed, enabling measurements of both hourly radiance and daily irradiance. Land/sea mask, snow cover, angular components (solar and viewing geometry), and terrain height data served as additional auxiliary data within GEMS L1C. Furthermore, for atmospheric correction purposes, three GEMS L2 products were utilized: cloud (CLD) (Kim et al., 2024), aerosol (AERAOD) (Cho et al., 2023), and total column ozone (TCO) (Baek et al., 2023).

CLD data from GEMS were employed to identify and exclude cloudy pixels during the atmospheric correction process. The

GEMS CLD output is an optical quantity observed at UV-VIS wavelengths, which may differ from the physical properties of real clouds; therefore, it does not provide an official cloud mask. However, if the effective cloud fraction (ECF) is less than 0.2 and the cloud centroid pressure (CCP) is equal to 1013 hPa, the quality flag in the CLD output indicates a clear sky. However, to mitigate potential errors caused by lower clouds being mistaken as clear-sky, an additional cloud-masking step was performed when the CCP was equal to or greater than 1000 hPa, following the initial masking based on the ECF.

The GEMS AERAOD data provided AOD values for three wavelengths (354, 443, and 550 nm). The GEMS AOD at 443 nm shows high accuracy with a strong positive correlation coefficient (R-value) of about 0.89 and a low root mean squared error (RMSE) of 0.15 after validation with the AERONET observed AOD (Cho et al., 2023). In this study, we utilized 550 nm AOD data to perform atmospheric correction.

The GEMS TCO output was used to access the entire column of ozone data for atmospheric calibration. GEMS TCO showed

a high correlation coefficient (R-value) of 0.97 and a low RMSE of 1.3 Dobson Unit (DU) when compared to Pandora TCO measurements, which is approximately the same or better than comparisons with the ozone mapping and profiler suite (OMPS) and TROPOMI (Baek et al., 2023).





### 2.3 Copernicus Atmosphere Monitoring Service (CAMS) near-real-time data

Surface reflectance can only be calculated for pixels that are classified as clear in the CLD data and when both AOD and TCO
data are available. However, the processing method for pixel quality in the GEMS AOD output differs from that of other products,
which can result in missing calculations in certain areas, even in clear skies. To address this issue, we used additional AOD from
the Copernicus Atmospheric Monitoring Service (CAMS). The CAMS AOD data were available in a daily, 0.25-degree grid
format provided by the European Center for Medium-Range Weather Forecasts (ECMWF). When a pixel was classified as clear
in the CLD algorithm but lacked in GEMS AOD data, interpolation was performed using the CAMS AOD.

### 2.4 Radiometric Calibration Network (RadCalNet) ground measurement data: For validation

RadCalNet, an acronym for radiometric calibration networks, serves as a pivotal resource for Earth observation satellites by
measuring and furnishing land surface reflectance values at five strategically positioned measurement points worldwide (Bouvet
et al., 2019). The primary objective of RadCalNet is to facilitate the calibration process, and it has already been proven to be
instrumental in validating ground reflectance data for prominent satellites, such as Landsat (Voskanian et al., 2023) and Sentinel
(Gao et al., 2021). The land surface reflectance data provided by RadCalNet encompassed suface reflectance observed at 10 nm
wavelength intervals ranging from 400 nm to 2500 nm at 30-min intervals spanning from 01:00 to 07:00 UTC. Among the five
designated measurement sites, Baotou (BTCN) and Baotou Sand (BSCN) are located in the Baotou region of China and fall
within the coverage area of the GEMS satellite. Because the BTCN is primarily used for calibrating high-resolution optical
satellite imagery, the BSCN, which is located in a desert area, was used for validation in this study. Consequently, for this study,
data from the BSCN site were meticulously collected and employed to verify ground reflectance, contributing to the robustness
of the research findings.

### 3 Background Surface Refelctance (BSR) retrieval algorithm

Figure 1 depicts a comprehensive flow chart of the BSR retrieval algorithm, which comprises two primary steps: (1) atmospheric
correction and (2) BRDF modeling and BSR retrieval. Subsequently, the constructed GEMS BSR and LER were validated
through comparative analysis with the GEMS TOC in this study.The methodology and underlying assumptions are detailed in
the subsequent subsections.

### 3.1 Atmospheric correction

Atmospheric correction plays a pivotal role in remote sensing by rectifying distortions caused by atmospheric effects, which can
vary based on different geometries and atmospheric conditions. These atmospheric effects introduce significant uncertainties
when the Earth's surface is observed using satellite imagery. Thus, atmospheric correction is crucial for accurately determining
surface reflectance. In most satellite data-processing methods, atmospheric correction is achieved using Radiative Transfer
Models (RTMs). top-of-atmosphere (TOA) reflectance can be calculated using the atmospheric correction coefficients derived





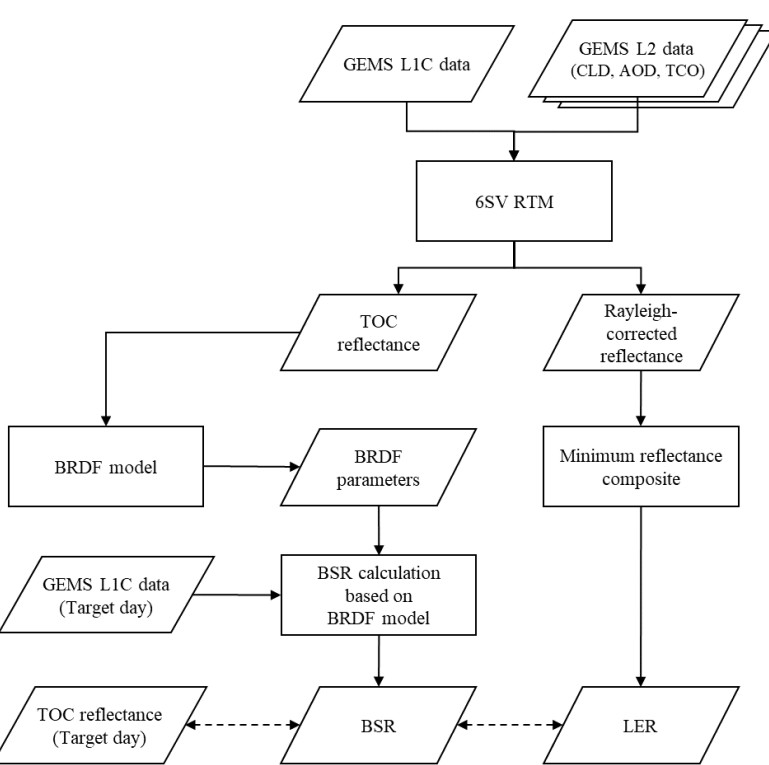

**Figure 1.** Flowchart of this study

from the RTM, it is possible to calculate the top-of-canopy (TOC) reflectance from the TOA reflectance. The Second Simulation of the Satellite Signal in the Solar Spectrum Vector (6SV) RTM (Vermote et al., 2006) was employed for atmospheric correction, which was utilized in MODIS, the Visible Infrared Imaging Radiometer Suite (VIIRS) (Roger et al., 2016), and the Geostationary Operational Environmental Satellite (GOES) (Peng, 2020) data. The 6SV RTM divides the wavelength into intervals of 2.5 nm and computes the scattering and absorption effects in the atmosphere caused by aerosols and their geometric components. By employing the three atmospheric correction coefficients (xap, xb, and xc) derived from the 6SV RTM, the TOC reflectance can be calculated from the TOA reflectance. The surface reflectance was then calculated from the TOA reflectance, as expressed in Equation (1). The atmospheric correction coefficients xap, xb, and xc can be computed using Equations (2), (3), and (4), where they represent the inverse of the transmittance, scattering term of the atmosphere, and spherical albedo, respectively. In these equations, $\rho_s(\theta_s, \theta_v, \phi)$, $\rho_A(\theta_s, \theta_v, \phi)$, $\theta_s$, $\theta_v$, $\phi$, $T^{\uparrow}(\theta_s)$, $T^{\downarrow}(\theta_v)$, $T_g$, and $S$ denote TOA reflectance, TOC reflectance, the Solar Zenith Angle (SZA), Viewing Zenith Angle (VZA), and Relative Azimuth Angle (RAA), atmospheric transmittance (sun to target), atmospheric transmittance (target to sun), total gas transmittance, and spherical albedo, respectively.





$$\rho_s(\theta_s, \theta_v, \phi) = \frac{xap \cdot \rho_A(\theta_s, \theta_v, \phi) - xb}{1 + xc(xap \cdot \rho_A(\theta_s, \theta_v, \phi) - xb)} \tag{1}$$

$$xap = \frac{1}{T_g(\theta_s, \theta_v)T^\uparrow(\theta_s)T^\downarrow(\theta_v)} \tag{2}$$

$$xb = \frac{\rho_a(\theta_s, \theta_v, \phi)}{T^\uparrow(\theta_s)T^\downarrow(\theta_v)} \tag{3}$$

$$xc = S \tag{4}$$

Although the RTM offers high accuracy in surface reflectance calculations, its computational complexity and time requirements are significant. To address this challenge, surface reflectance algorithms often use pre-simulated lookup tables (LUTs) generated using RTMs under various conditions. During LUT configuration, careful selection of input variables is crucial, as they directly impact the accuracy of atmospheric correction and surface reflectance calculations. In alignment with the characteristics of the GEMS, six input variables were chosen for LUT construction: SZA, VZA, RAA, TCO, AOD, and terrain height. Table 1 outlines the range and intervals of these input variables for LUT construction with reference to previous studies on LUT-based surface reflectance calculations (Peng, 2020; Lee et al., 2020). Nevertheless, even with detailed interval adjustments, rapid changes at high angles can lead to discontinuities and degradation of the surface reflectance. Therefore, to ensure smooth and accurate results when calculating the surface reflectance based on the LUT, real-time interpolation was performed for all input components (SZA, VZA, RAA, TCO, AOD, and terrain height).

**Table 1.** 6SV-based LUT input variables and its interval.

| Input parameter (unit) | Min | Max | Increment |
|---|---|---|---|
| SZA (degree) | 0 | 80 | 0-70(5), 70-80(2) |
| VZA (degree) | 0 | 80 | 5 |
| RAA (degree) | 0 | 180 | 10 |
| TCO (DU) | 250 | 350 | 50 |
| Terrain height (km) | 0 | 3.5 | 0.5 |
| AOD | 0.01, 0.05, 0.1, 0.15, 0.2, 0.3, 0.4, 0.6, 0.8, 1.0, 1.5 | | |
| Aerosol type | Continental | | |





### 3.2 BSR retrieval through BRDF modeling

The computation of surface reflectance through atmospheric correction presents a significant limitation: its effectiveness is restricted to cloud-free regions and is influenced by the observational geometry at the time of measurement. Consequently, there is a critical need for a methodology capable of simulating the surface reflectance across diverse angular conditions to accurately compute the BSR. Thus, prior studies that required the simulation of surface reflectance at various angles, such as albedo calculations, have characterized the anisotropic properties of surfaces by utilizing the BRDF model (Gao et al., 2003;

Wen et al., 2018). BRDF models can be classified into three categories: physical, empirical, and semi-empirical. Although a physical model can express the inherent physical meaning of each parameter mathematically, its versatility is limited because of its computationally intensive nature. The empirical model utilizes observation-based empirical formulas, making it suitable for situations with limited observations; however, it requires a substantial number of observations and does not elucidate the underlying physical background. Consequently, semi-empirical BRDF models are widely employed in remote sensing.

The computation of surface reflectance through atmospheric correction presents a significant limitation: its effectiveness is restricted to cloud-free regions and is influenced by the observational geometry at the time of measurement. Consequently, there is a critical need for a methodology capable of simulating surface reflectance across diverse angular conditions to accurately compute BSR. Thus, prior studies that require the process of simulating surface reflectance at various angles, such as albedo calculation, have characterized the anisotropic properties of surfaces by utilizing the BRDF model (Gao et al., 2003; Wen

et al., 2018). The BRDF model can be classified into three categories: physical, empirical, and semi-empirical. While the physical model can express the inherent physical meaning of each parameter mathematically, it is limited in versatility due to its computationally intensive nature. The empirical model utilizes observation-based empirical formulas, making it suitable for situations with limited observations, but it requires a substantial number of observations and does not elucidate the underlying physical background. Consequently, in the realm of remote sensing, the semi-empirical BRDF model is widely employed.

In this algorithm, the semi-empirical Roujean Bidirectional Reflectance Distribution Function (BRDF) model (Roujean et al., 1992) was utilized for BRDF modeling. The Roujean BRDF model defines surface reflectance as a combination of isotropic, geometric, and volumetric scattering components. It comprises two physical kernels (f1 and f2) and three empirical coefficients (K0, K1, K2; BRDF parameters) that describe the mechanism of each component, as shown in Equation 5. The two physical kernels of the Roujean BRDF model were defined under the assumption of irregularly spaced rectangular protrusions on a flat

ground surface, neglecting all shadow interactions. The two physical kernels are calculated based on the angular component at the time of the observation(formulas are detailed in (Roujean et al., 1992)).

$$R(\theta_s, \theta_v, \phi) = K_0 + K_1 \cdot f1(\theta_s, \theta_v, \phi) + K_2 \cdot f2(\theta_s, \theta_v, \phi) \tag{5}$$

The observed surface reflectance can be decomposed into two kernels and three variables, as described by Equation 5. Since the two kernels can be computed from the angular components, the BRDF parameters which characterize the anisotropic

reflection based on multiple observations collected over a synthesis period can be calculated using the method of least squares.This method assumes a relatively stable anisotropic reflection behavior of the surface over short periods in the absence




of significant disturbances such as forest fires or floods. Therefore, these variables can be applied within a short timeframe, enabling the simulation of surface reflectance under clear-sky conditions across all angular configurations within the target scene. Consequently, once the BRDF parameters are computed, by providing solely on the angular component of the observed

target scene, the BSR at any angle can be simulated.

In this algorithm, the "target scene" refers to the day following the completion of BRDF synthesis. Figure 2 illustrates the synthesis period of 15 d and the retrieval cycle used for BRDF modeling in the algorithm. The BRDF synthesis period spans 15 days and encompasses all observed hourly data within this timeframe. This enabled the resulting BRDF composites to be applied to the subsequent day, allowing for the simulation of hourly BSR based solely on the angular components observed the

following day. These BRDFs were generated in a daily cycle, with the synthesis period shifted by one day. This methodology enables comprehensive reconstruction of the surface reflection characteristics at various angles, offering valuable insights into the directional reflection behavior of the surface.

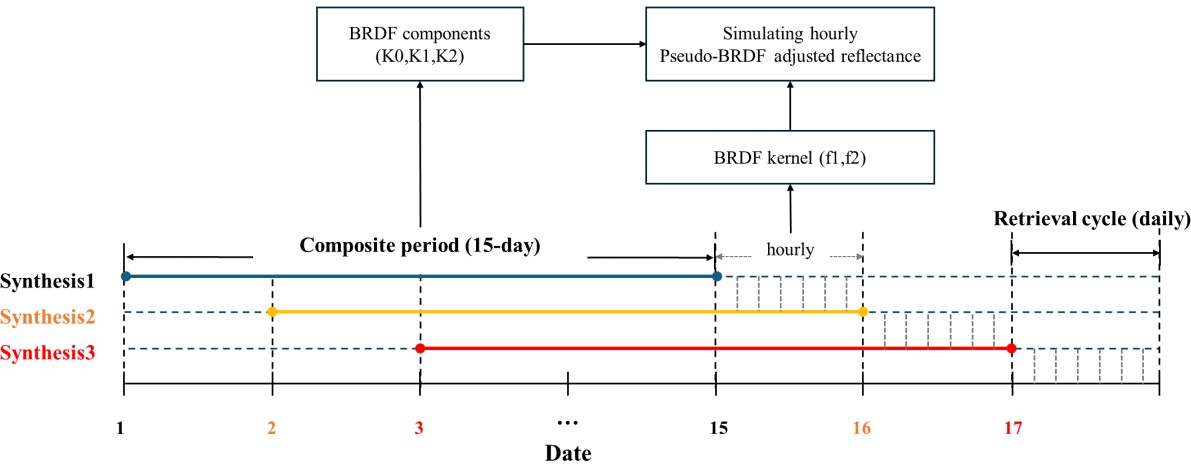

**Figure 2.** Schematic of 15-day composite period and retrieval cycle for BRDF modeling

Insufficient observations lead to uncalculated BRDF parameters, resulting in missing BSR values, which are crucial for other L2 algorithms. To address this, a gap-filling process was implemented, utilizing the "Age" variable in BRDF modeling. This

method uses previously calculated BRDF parameters up to 5 days of age to fill gaps when current parameters are unavailable. The Age variable indicates the number of days since the last valid BRDF calculation, reset to 0 after 5 days, or upon the BRDF parameter calculation.

## 3.3 GEMS LER generation

The GEMS LER was derived from the Rayleigh-corrected reflectance, which was computed by assuming a Rayleigh atmosphere

to eliminate aerosol effects during atmospheric correction. This calculation utilized the same LUT as the TOC reflectance calculation with the AOD set to zero. It was assumed that the formulas for calculating the LER and 6SV-based atmospheric





corrections were nearly identical. Equations 6 and 7 summarize these formulas, where $\rho_{6SV}$ and $\rho_{LER}$ represent the 6SV-based TOC and LER calculation formulas (Kleipool et al., 2008), respectively. In this study, $\rho_{6SV}$ was designated as the Rayleigh-corrected reflectance for the scene. The "GEMS LER" was determined as the minimum reflectance over a 15-day period, aligning

with the BRDF synthesis period. Additionally, the GEMS LER was used for further gap filling in cases of persistent missing in GEMS BSR, despite utilizing the age variable.

$$\rho_{6sv} = \frac{\rho'_s}{1 + \rho'_s S} \quad \text{with} \quad \rho'_s = \frac{\frac{\rho_A(\theta_s, \theta_v, \theta_\phi)}{T_g} - \rho_a(\theta_s, \theta_v, \theta_\phi)}{T^\downarrow(\theta_s) T^\uparrow(\theta_v)} \tag{6}$$

$$\rho_{LER} = \frac{\rho_A(\theta_s, \theta_v, \theta_\phi) - \rho_a(\theta_s, \theta_v, \theta_\phi)}{T^\downarrow(\theta_s) T^\uparrow(\theta_v) + S(\rho_A(\theta_s, \theta_v, \theta_\phi) - \rho_a(\theta_s, \theta_v, \theta_\phi))} \tag{7}$$

## 4   Results

### 4.1   BSR validation with TOC reflectance

#### 4.1.1   Quantitative validation of GEMS BSR and LER

The purpose of the BSR data was to simulate reflectance as similar as possible to the TOC reflectance calculated based on actual observation conditions in advance and use it as an input for other L2 outputs. Therefore, verification was performed by comparing the SFC calculated after the L2 outputs (CLD, AOD, and TCO) utilized for the TOC reflectance calculation were

produced with the BSR simulated in advance as a reference value. We have produced TOC reflectance and BSR data for one year, from January 1 to December 31, 2021, and performed BSR verification based on these data.

The BSR validation was limited to instances of high quality, with the quality criteria defined based on the number of observations and RMSE during the BRDF modeling. The number of observations refers to the number of valid pixels used within the same pixel for BRDF modeling, whereas the BRDF RMSE signifies the RMSE between the actual observed reflectance

and simulated reflectance. In this context, the simulated reflectance represents the value that emulates the reflectance back to the angular component of the actual observation condition based on the BRDF parameters derived through BRDF modeling. The GK-2A AMI albedo output establishes the standard for good quality BSR, requiring seven or more observations and a BRDF RMSE of 0.07 or lower (Lee et al., 2020). In addition, MODIS Albedo considers an RMSE value within 10% of the channel-specific reflectance distribution as indicative of good quality (Román et al., 2013). Consequently, the BSR quality

criterion in this study was defined as requiring seven or more observations and a BRDF RMSE of 0.03 or lower, given that the GEMS 440 nm BSR typically reaches a maximum value of 0.3. All subsequent quantitative analyses were conducted exclusively using good-quality data.

Figure 3 presents a comparison between GEMS BSR and GEMS LER using the GEMS TOC reflectance as a reference, encompassing all hourly data from January to December 2021. The comparison revealed that the GEMS BSR simulated the



TOC values more closely than the GEMS LER. To quantify this comparison, we evaluated the RMSE, relative-RMSE (rRMSE), and bias. The rRMSE was computed by dividing the RMSE by the average reference data value, which served as an indicator of overall relative accuracy. For the GEMS BSR, the RMSE was 0.015, rRMSE was 19.38%, and bias was 0.002. Conversely, for GEMS LER, the RMSE was 0.017, the rRMSE was 22.17%, and the bias was -0.009. These results suggest that the GEMS BSR exhibited a lower rRMSE of 3% and a lower bias value of 0.007 (based on the absolute value), indicating superior simulation

performance compared to the minimum reflectance.

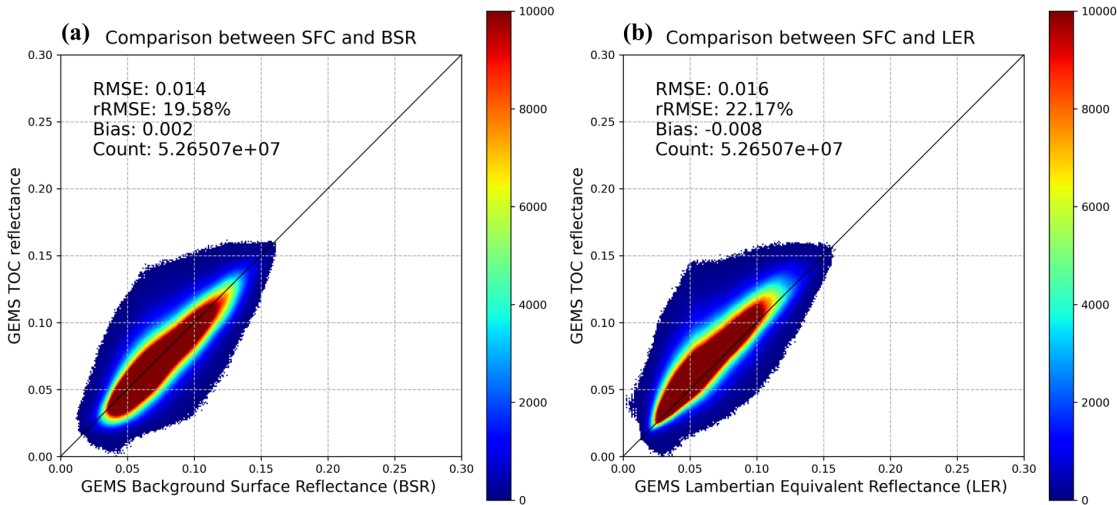

**Figure 3.** Assessment of GEMS BSR and LER accuracy relative to GEMS TOC reflectance

Table 2 presents a quantitative comparison between GEMS BSR and LER for each season. The seasons are represented as DJF, MAM, JJA, and SON, corresponding to December-January-February, March-April-May, June-July-August, and September-October-November, respectively, and denote winter, spring, summer, and fall, respectively. The RMSE, rRMSE, and bias values for BSR are 0.012, 0.016, 0.017, and 0.012 for RMSE; 18.89, 20.5, 21.73 and 18.19% for rRMSE; and -0.002, 0.007, 0.004,

and -0.002 for bias, respectively, in the order of winter, spring, summer, and fall. For the GEMS LER, the RMSE values were 0.013, 0.017, 0.019, and 0.013, respectively; the rRMSE values were 21.9, 20.85, 24.16, and 20.08%, respectively; and the bias values were -0.006, -0.009, -0.009, and -0.007 for each season, respectively. BSR exhibited lower RMSE, rRMSE, and bias values than LER, indicating the superior simulation performance of BSR in all seasons. Both data sources show the highest RMSE in summer compared to the other seasons. However, the difference in rRMSE compared to RMSE did not change much

across the seasons because the reflectance in winter and fall was relatively lower than that spring and summer.

### 4.1.2 Qualitative validation of GEMS BSR and LER

After completing the quantitative analysis, we conducted a qualitative comparison between the GEMS TOC, BSR, and LER to evaluate their spatial distribution. Figure 4 illustrates this qualitative analysis, with each row displaying the GEMS SFC,





**Table 2.** Seasonal quantitative comparison of GEMS BSR and LER with GEMS TOC reflectance as reference (DJF means December-January-February, MAM means March-April-May, JJA means June-July-August and SON means September-October-November)

| period | DJF | | MAM | | JJA | | SON | |
|---|---|---|---|---|---|---|---|---|
| Reflectance data | BSR | LER | BSR | LER | BSR | LER | BSR | LER |
| RMSE | 0.012 | 0.013 | 0.016 | 0.017 | 0.017 | 0.019 | 0.012 | 0.013 |
| rRMSE(%) | 18.89 | 21.9 | 20.5 | 20.85 | 21.73 | 24.16 | 18.19 | 20.08 |
| Bias | -0.002 | -0.006 | 0.007 | -0.009 | 0.004 | -0.009 | -0.002 | -0.007 |

BSR, LER, and the difference between BSR and LER relative to SFC for the same date and time. Subfigures (a)–(d) in Figure 4
represent the calculations for February 20 at 0245 UTC, May 1 at 0345 UTC, August 9 at 0245 UTC, and November 10, 2021, at 0545 UTC.

TOC reflectance is applicable only under clear-sky conditions and can be computed only in areas where all necessary inputs, such as AOD and TCO, are available, resulting in numerous areas with missing data. The difference between the BSR and LER data in terms of TOC was examined only for areas where the GEMS TOC was calculated. For BSR, a mixed trend of
underestimation and overestimation was observed; however, in July, most areas exhibited higher reflectance than TOC. In contrast, the LER showed a consistent trend of underestimation in most areas, except for October, which aligns with our expectations. However, in October, LER exhibited a similar trend to BSR, but with a stronger magnitude, indicating that even applying minimum reflectance can result in higher values than those from BRDF modeling. This again highlights the importance of considering the BRDF when calculating land surface reflectance.

A qualitative analysis was also conducted between the BSR and LER for areas where TOC reflectance was not calculated. The greatest difference between the two sources was observed in July (summer), compared with the spring, summer, and fall results in February, April, and October, respectively. In July, the LER consistently simulated lower reflectivity than the BSR across the entire region, particularly in the central and eastern parts of the country. These areas experience frequent cloud cover and fog throughout the year, making it challenging to obtain clear-sky pixel counts during all seasons. However, this
challenge is exacerbated in summer months when clouds are more prevalent. If clouds are not detected in the cloud outputs, high reflectance can be inputted into the BRDF model and mistakenly interpreted as clear-sky reflectance, leading to simulation errors. Conversely, the minimum reflectance technique tends to adopt the lowest reflectance value during the synthesis period, even in the presence of clouds and shadows, resulting in a lower reflectance.



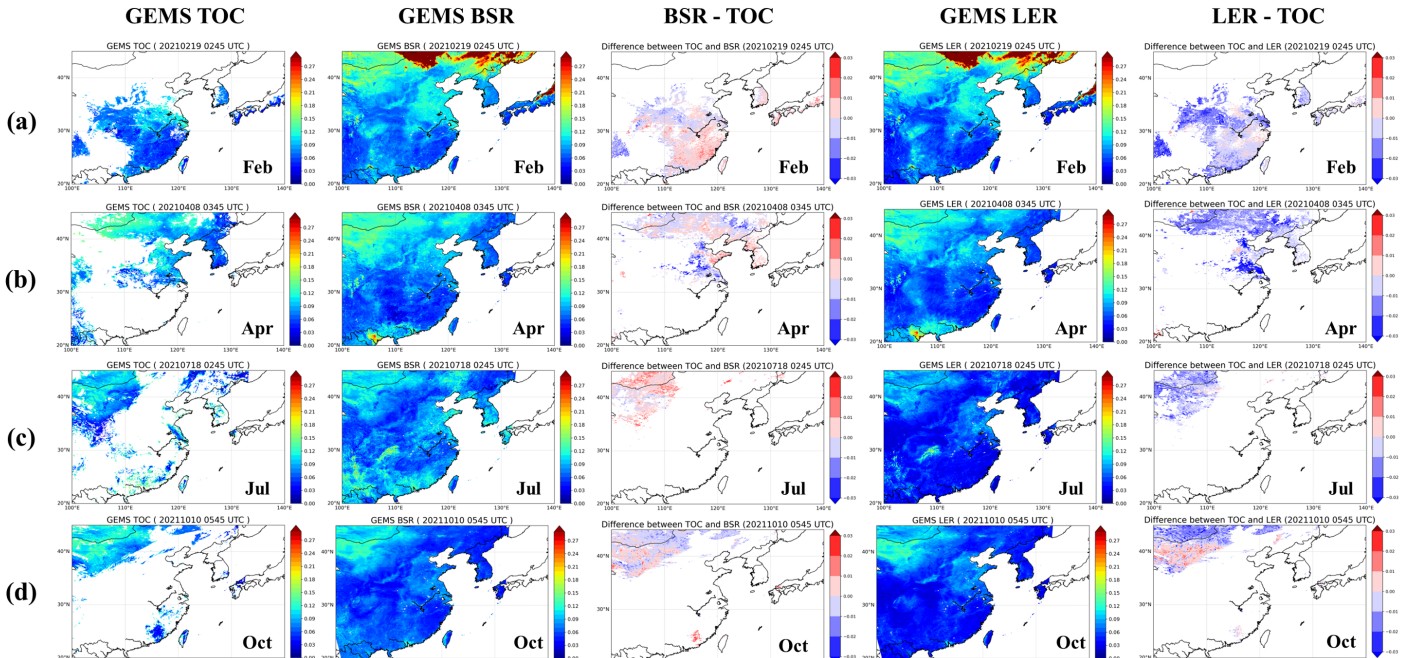

**Figure 4.** Qualitative comparison of GEMS TOC, BSR, and LER. (a) GEMS TOC; (b) GEMS BSR; (c) difference between BSR and TOC; (d) GEMS LER; (e) difference between LER and TOC.

## 4.2 Analyzing surface reflectance variation across land types

### 4.2.1 Time series consistency analysis by land types

To assess the simulation performance based on the time series of the BSR, we analyzed the time series stability across four land types (grassland, cropland, shrubland, and urban) using MODIS land cover data. Figure 5 illustrates the time series of GEMS TOC, BSR, and LER analyzed for each land type, where subfigures (a)–(d) correspond to grassland, cropland, shrubland, and urban areas, respectively. These values were averaged over all good-quality pixels for each land type based on MODIS land cover. In this analysis, the LER was excluded if it did not meet the criteria for good-quality BSR to mitigate the impact of clouds.

The blue squares in Figure 5 represent the reference TOC values, green circles represent the LER values, and red circles represent the BSR values. The analysis indicated that for all four land types, both BSR and LER exhibited stable time-series distributions, mirroring the TOC trend. However, when compared with the TOC distribution, the BSR appeared to track the trend slightly better than the LER across all land types. This was most evident between May and October, as LER consistently simulated a lower reflectance than TOC. In addition, in the shrublands during this period, LER tended to consistently adopt nearly identical values, whereas TOC and BSR showed variability in reflectance.

However, for grassland, the trend was opposite to that of LER, with BSR consistently simulating a higher reflectance than TOC during the summer months. While there was a clear tendency for the BSR to simulate higher reflectance, this was similar



to the observed range of the TOC reflectance distribution and was not considered a significant error. Therefore, this analysis

confirms that the BSR tends to follow the TOC reflectance more reliably than the LER over time.

**Figure 5.** Time series distribution of TOC, BSR, and LER by land type. (a) Grassland; (b) Cropland; (c) Shrubland; (d) Urban





### 4.2.2 Surface reflectance influence on AOD variability in cropland and urban areas

(Li et al., 2012) provided quantitative figures for the variability of the AOD output with reflectance changes in the blue channel. The analysis focused on different AOD ranges (AOD<0.4, 0.4<AOD<0.8, 0.8<AOD<1.2, 1.2<AOD<1.6, 1.6<AOD<2.0) according to the degree of ground reflectance change (ranging from 0.001 to 0.05). To quantitatively evaluate the advantage of

BSR over LER in terms of AOD, we reanalyzed the results of (Li et al., 2012) using the findings of this study. The analysis was limited to the AOD value is 0.4, and the variability was analyzed and converted into percent error using a quadratic regression equation based on the change in AOD value corresponding to the change in ground surface reflectance as presented in the study. In this context, the amount of AOD change refers to the difference in AOD values when using BSR and LER data compared to when using TOC, assuming that the AOD output value when using TOC is 0.4. The percent error of AOD can be calculated using

Equation 8, where $AOD_{reference}$ is the AOD value when using TOC (AOD is fixed at 0.4 in this study) and $AOD_{observed}$ is the AOD value when using BSR and LER. However, as Li's study only analyzed urban and rural areas in China, it also focused on cropland and urban land types.

$$Percenterror_{AOD} = \frac{AOD_{observed} - AOD_{reference}}{AOD_{reference}} * 100 \tag{8}$$

Figure 6 presents the seasonal RMSE, bias, and AOD percent error results, with Figures 6(a) and 6(b) representing the results

for cropland and urban areas, respectively. In each figure, the first to third columns denote the RMSE, bias, and AOD percentage error based on TOC, respectively. Both RMSE and bias values were expressed by multiplying the calculated value by 100. Throughout all seasons, BSR consistently exhibited lower RMSE and bias values than LER. LER tended to show a negative bias, whereas BSR showed a positive bias. Interestingly, both datasets demonstrated high RMSE values in summer, with the BSR tending to overestimate and the LER tending to underestimate. This inverse behavior is in line with the trends observed in the

previous time-series graphs. Specifically, in the cropland and urban areas, the BSR maintained a low bias of 0.0022 or less, except in summer. In contrast, LER exhibited a negative bias of more than 0.084 in most cases, except in winter in the croplands.

Analysis of the AOD percent error revealed that LER exhibited higher error values than BSR across all seasons. Due to the high RMSE values observed in summer for both datasets, the AOD percentage error also tended to increase during this season. However, excluding summer, BSR consistently demonstrated better AOD percentage error values than LER, with improvements

of up to 7.4% in spring for croplands, and 10.9% in winter for urban areas. These findings suggest that the BSR has the potential to generate more stable AOD outputs than the LER.





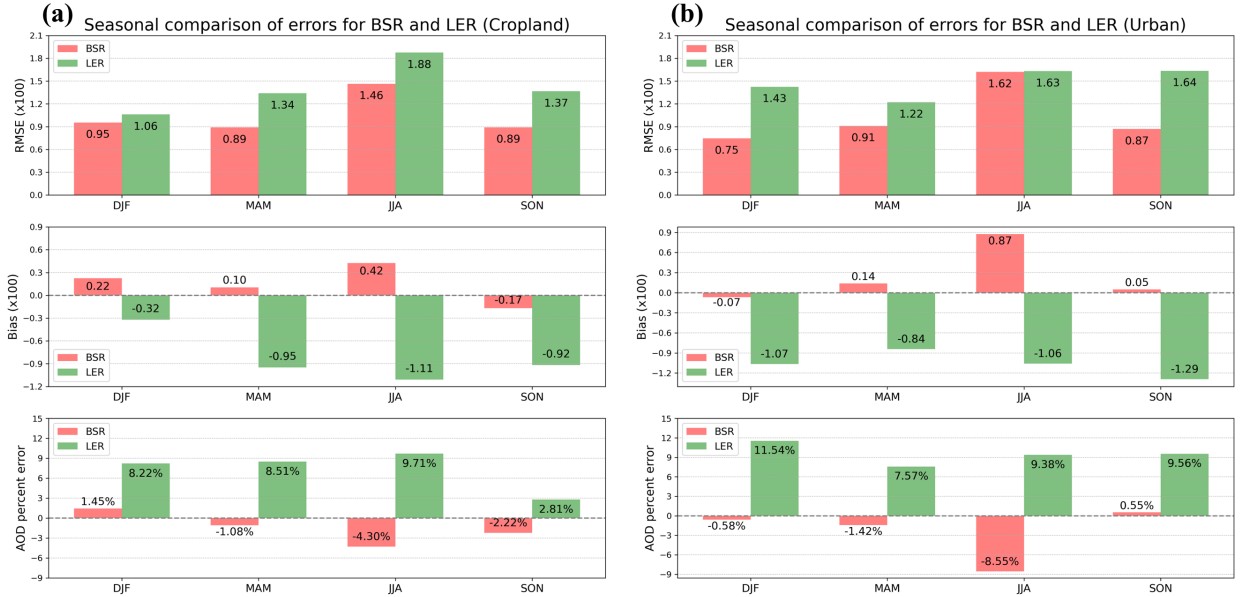

**Figure 6.** RMSE and bias in cropland and urban by season, and the resulting AOD percent error. (a) Cropland; (b) Urban

## 4.3 Accuracy evaluation using ground measurements

To conduct a thorough and precise validation of GEMS BSR, comparisons and validations were performed using RadCalNet ground observations. RadCalNet data utilized ground reflectance observations from the BSCN sites, and only data within 15 min

of the observation time were used because of the sensitivity of ground reflectance to temporal changes. In addition, various LER databases currently available in the field were used for verification to assess the performance of the BSR. These include the OMI LER, GLER, TROPOMI DLER, GOME-2 LER, and SCIAMACHY LER. OMI GLER and DLER considered surface BRDF effects and conducted observations over the study area, mostly between 03:00 and 05:00 UTC. Therefore, the validation was limited to data from 03:45-05:45 UTC based on RadCalNet. For TROPOMI DLER, the BRDF effect on the viewing geometry

can be accounted for using four coefficients. These coefficients can be derived from the LER and converted into the DLER using the following formula 9: For comparison and validation, the GEMS VZA at 0445 UTC was multiplied by the corresponding coefficients to calculate the TROPOMI DLER. $\theta_v$ in the formula is the GEMS VZA and from $c_0$ to $c_4$ are the TROPOMI DLER coefficients.

$$\rho_{DLER} = \rho_{LER} + c_0 + c_1 \cdot \theta_v + c_2 \cdot \theta_v^2 + c_3 \cdot \theta_v^3 \tag{9}$$

Figure 7 illustrates the results of the validation of the LER databases, including the GEMS BSR, based on RadCalNet observations. Figure 7(a) compares OMI GLER, TROPOMI DLER, and GEMS BSR with the RadCalNet data, whereas Figure 7(b) compares OMI, GOME-2, SCIAMACHY LER, and GEMS BSR. The RMSE, rRMSE, and bias between each dataset and RadCalNet are presented in Table 3. The analysis revealed that the RMSE of OMI GLER, TROPOMI DLER, and GEMS BSR





were 0.018, 0.009, and 0.007, respectively, with rRMSE values of 21.42, 10.57, and 8.59%, and bias values of -0.008, -0.006,
and 0.001, respectively. These results indicate that the GEMS BSR is more accurate than the OMI GLER and TROPOMI DLER
in terms of RMSE, rRMSE, and bias, with an 13% improvement over the OMI GLER and 2% improvement over the TROPOMI
DLER based on the rRMSE.

Furthermore, OMI GLER exhibited a significantly wider distribution than the RadCalNet reflectance occurrence range,
whereas TROPOMI DLER had almost the same reflectance adopted multiple times. This suggests that similar DLER values
were calculated multiple times in the same month because of the nature of the LER database data, as the GEMS satellite is
geostationary and the VZA varies little at the same observation time. Similar trends were observed when comparing the three
LER databases (OMI, GOME-2, and SCIAMACHY), with RMSE values of 0.025, 0.013, 0.014, rRMSE values of 29.82, 16.2,
16.81%, and bias values of -0.024, -0.012, -0.01, respectively.

In conclusion, considering BRDF effects, such as BSR, GLER, and DLER, rather than utilizing LER databases for alternative
land surface reflectance calculations, can provide more realistic reflectance simulations, with GEMS BSR demonstrating the
best performance among the six sources analyzed.

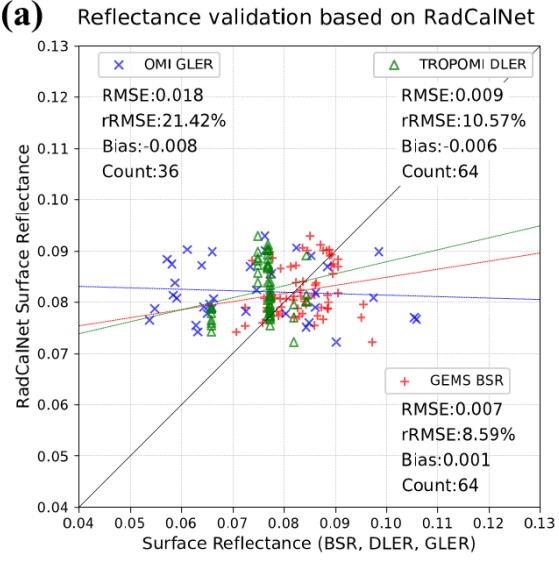

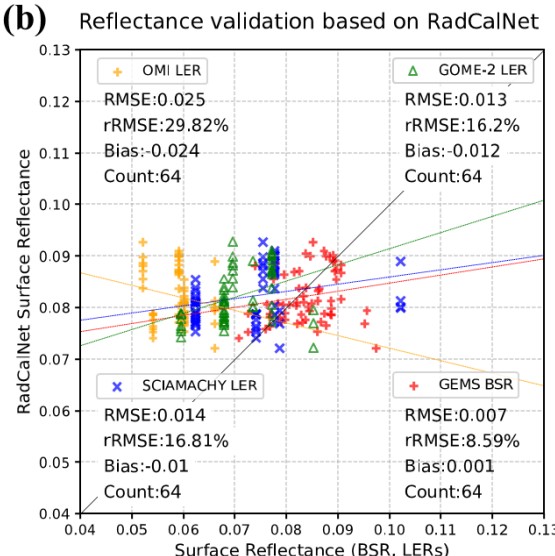

**Figure 7.** Validation of surface reflectance databases using RadCalNet observations. (a) Validation of OMI GLER, TROPOMI DLER and
GEMS BSR; (b) Validation of OMI LER, GOME-2 LER, SCIAMACHY LER, and GEMS BSR.





**Table 3.** GEMS BSR and LER database accuracy evaluation based on RadCalNet ground observation data

| Data | GEMS BSR | OMI GLER | TROPOMI DLER | OMI LER | GOME-2 LER | SCIAMACHY LER |
|------|----------|----------|--------------|---------|------------|---------------|
| RMSE | 0.007 | 0.018 | 0.009 | 0.025 | 0.013 | 0.014 |
| rRMSE(%) | 8.59 | 21.42 | 10.57 | 29.82 | 16.2 | 16.81 |
| Bias | 0.001 | -0.008 | -0.006 | -0.024 | -0.012 | -0.01 |

## 4.4  Intercomparison between GEMS BSR and LER database (GEMS LER, OMI GLER, TROPOMI DLER)

TROPOMI and OMI performed observations between 0345 and 0545 UTC during GEMS observations in the study area, with the largest number of observations occurring at 0445 UTC. Therefore, a comparison of the GEMS BSR with the DLER and GLER was conducted at 0445 UTC. Consistent with the analysis in the previous section, OMI GLER was calculated using only the data within 15 min of the GEMS observations, whereas TROPOMI DLER was calculated using GEMS VZA. Figure 8 compares GEMS TOC with GEMS BSR, TROPOMI DLER, and OMI GLER.

Compared to GEMS, TOC, BSR, DLER, and GLER were  17.78, 22.32, and 41.51% based on the rRMSE, with biases of 0.003, -0.002, and -0.025, respectively. The GEMS BSR tended to exhibit a positive bias, whereas the DLER and GLER tended to display a negative bias. When analyzing the graph distribution, we observed a significant clustering of values below 0.05 in the OMI GLER data. Owing to the inherent differences in assumptions between the GLER and DLER databases and BSR, direct quality verification through comparison is challenging. However, this suggests that the BSR is capable of simulating reflectivity that closely aligns with TOC reflectance, indicating its potential for realistic reflectivity simulation.

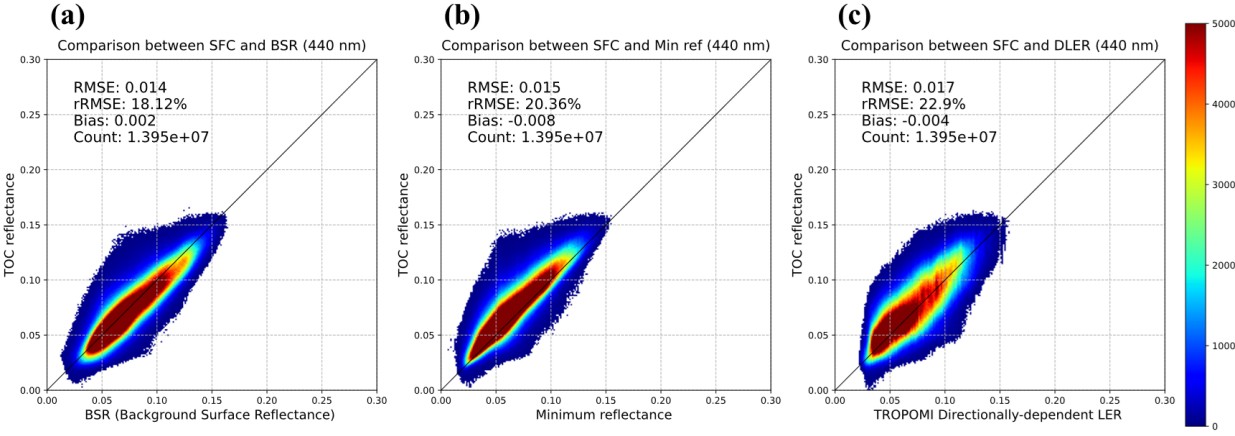

**Figure 8.** Distribution density plot of GEMS BSR, TROPOMI DLER, and OMI GLER based on GEMS TOC observations. (a) GEMS BSR; (b) TROPOMI DLER; (c) OMI GLER





Figure 9 shows a density plot and histogram comparing BSR with OMI GLER and TROPOMI DLER. Subfigures (a) and

(c) show a comparison with GLER, whereas (b) and (d) show a comparison with DLER. The correlation coefficient (R) for

GLER was 0.7, with a root mean square difference (RMSD) of 0.035 and a bias of 0.028. For DLER, R, RMSD, and bias were

0.81, 0.017, and 0.006, respectively, indicating a strong positive correlation for both data sources. The histograms reveal that the

distributions of the DLER and BSR values are quite similar, whereas GLER is more concentrated at low ground reflectance

than BSR. This suggests that BSR yields results that are more akin to those of DLER than those of GLER. Although the BSR

showed relatively high values when the GLER and DLER were near 0.05 in the density plot distribution, it was not considered a

significant error owing to its small magnitude.

**Figure 9.** Quantitative comparison of GEMS BSR, TROPOMI DLER, and OMI GLER value distributions (a) Density plot of BSR and GLER; (b) Density plot of BSR and DLER; (c) Histogram of BSR and GLER; (d) Histogram of BSR and DLER





Figure 10 presents a qualitative comparison of the reflectances of GEMS BSR, TROPOMI DLER, and OMI GLER. The columns represent BSR, DLER, and GLER from left to right, and rows (a)–(c) represent dates 2021.03.26, 2021.06.23, and 2021.09.23, respectively. From the qualitative comparison, we can see that BSR and DLER have fairly similar distributions of values, whereas GLER tends to produce relatively low reflectance values in China compared with the other two datasets. However, as mentioned in the previous analysis, the BSR simulates a slightly higher reflectance during the summer months, with its values being relatively higher than those of the other dates. However, all three datasets had very similar distributions, albeit with slight differences in their absolute values. This analysis provides further evidence that the BSR effectively simulates the reflectance properties of the ground surface, similar to the DLER and GLER.

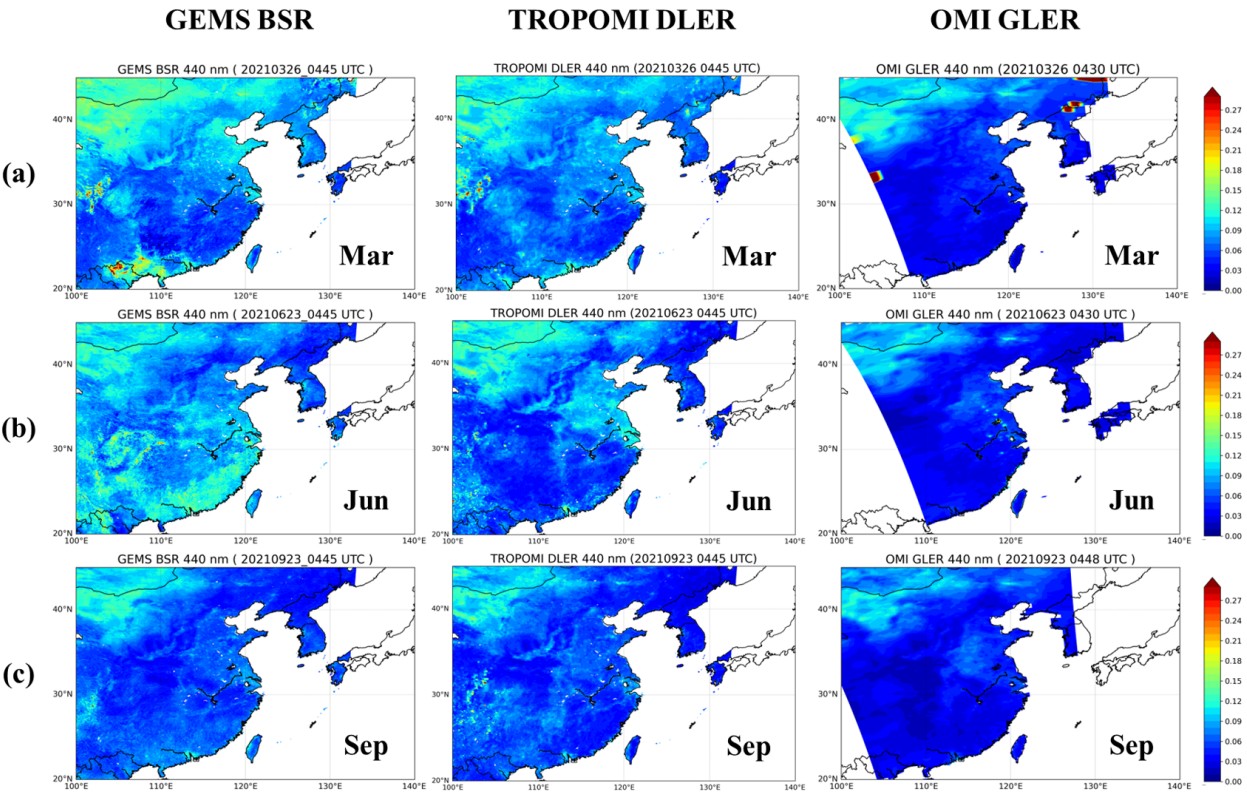

**Figure 10.** Qualitative comparison of GEMS BSR, TROPOMI DLER, and OMI GLER distributions in March, June, and September. The table rows correspond to GEMS BSR, TROPOMI DLER, and OMI GLER, in that order. (a) 2021.03.26; (b) 2021.06.23; (c) 2021.09.23

## 5 Conculusion

This study introduced the novel concept of BSR as an alternative output to resolve the output precedence dilemma between land surface reflectance and other L2 outputs applied to GEMS, a hyperspectral satellite observing in the UV-VIS range. This concept,



often referred to as BSR, involves a preliminary simulation of realistic ground reflectance before atmospheric correction. This simulation was based on variables reflecting the BRDF effect on the ground surface calculated through BRDF modeling. BSR
overcomes the limitation of underestimating ground reflectance that may occur with the minimum reflectance technique.

Various analytical methods were employed to evaluate the simulation performance of the BSR. The purpose of the BSR output was to simulate the TOC reflectance more realistically in advance using the actual observed values required for atmospheric corrections, such as CLD, AOD, and TCO. Therefore, TOC reflectance was used as a reference for evaluating the simulation performance of the BSR. The simulation performance of the GEMS BSR was 3% more accurate than that of the GEMS LER
data in terms of the rRMSE over the entire study period based on TOC. The bias was -0.007 for LER and 0.002 for BSR, indicating an improvement in the underestimation of surface reflectivity when BSR is applied.

The stability of the BSR calculation over time was also verified by analyzing the trend of the reflectance distribution according to MODIS land cover. For the selected terrains (grassland, agricultural, shrubland, and urban), the BSR tracked the TOC trend better than the LER. In summer, a tendency for BSR to be overestimated compared to TOC was observed in grasslands. In
contrast, LER was significantly underestimated compared to BSR in all land types, especially between May and September, confirming the temporal stability of the BSR.

Furthermore, after validation with TOC, comparison and validation were performed with other LER databases available in the field based on ground observations from RadCalNet. The OMI, GOME-2, SCIAMACHY LER, OMI GLER, and TROPOMI DLER data were used in the analysis. For the RadCalNet-based land surface reflectivity validation, GEMS BSR exhibited the
best simulation performance among the six databases, with an rRMSE of 8.59% and a bias of 0.001. When comparing the results of OMI GLER and LER validation, OMI GLER had an rRMSE of 21.42% and a bias of -0.008, while OMI LER had an rRMSE of 29.82% and a bias of -0.024. This indicates that accounting for surface BRDF effects in the calculation of ground reflectance provides a more realistic reflectance simulation.

A comparison of GEMS BSR with TROPOMI DLER and OMI GLER based on TOC reflectance revealed that BSR tended to
exhibit a positive bias, whereas DLER and GLER tended to display a negative bias. Despite the challenges in directly verifying the quality, the analysis suggests that BSR can simulate reflectivity closely aligned with TOC reflectance, indicating its potential for realistic reflectivity simulation. Additionally, the comparison between DLER and GLER based on BSR showed strong positive correlations, with BSR exhibiting distributions more similar to those of DLER. Overall, the BSR demonstrated the capability to effectively simulate ground surface reflectance properties, similar to the DLER and GLER.
In conclusion, our study demonstrated that BSR can effectively simulate realistic reflectance, surpassing the minimum reflectance approach used in many existing studies. Although limitations exist, such as the challenge of capturing sudden changes in surface characteristics such as snow or ice cover, our research is pioneering in its application to BRDF modeling and evaluation in UV-VIS observation satellite studies. By combining the high temporal and spatial resolution of GLER with the BRDF considerations of DLER, we laid the foundation for improved accuracy in the AQ output. Our findings suggest that the
utilization of BSR, a dataset reflecting realistic reflectivity with BRDF effects, can enhance various climate analysis studies, marking a significant advancement in the field.



*Data availability.* The data used in this study are accessible from the links below. GEMS Level-2 data (AERAOD, TCO, CLD) (https://nesc.nier.go.kr/en/html/datasvc/index.do); OMI/Aura Global Geometry-dependent surface LER (GLER)(https://disc.gsfc.nasa.gov/datasets/OMGLER_003/summary?keywords=AURAMLS); TROPOMI directionally dependent Lambertian-equivalent reflectivity (DLER) (https://www.temis.nl/surface/albedo/tropomi_ler.php); OMI surface LER database (https://www.temis.nl/surface/albedo/omi_ler.php); SCIAMACHY surface LER database (https://www.temis.nl/surface/albedo/scia_ler.php); GOME-2 surface LER database (https://www.temis.nl/surface/albedo/gome2_ler.php); CAMS global atmospheric composition forecasts dataset (https://ads.atmosphere.copernicus.eu/cdsapp#!/dataset/cams-global-atmospheric-composition-forecasts?tab=form); RadCalNet data (https://www.radcalnet.org/#!/)

*Author contributions.* S.S., and K-S.H. conceptualized and designed the study, collected and analyzed the data, and wrote the manuscript. S.C. and D.J. contributed to the study design, conducted the statistical analysis, and critically reviewed the manuscript. J.W., and H.K. assisted in data interpretation and manuscript preparation. N.K., and S.P., provided expertise in experimental procedures and contributed to the data collection. U.J., and H.H. contributed to the study design and provided intellectual input throughout the research process. All authors read and approved the final manuscript.

*Competing interests.* No potential conflict of interest was reported by the authors.

*Special issue statement.* This article is part of the special issue "GEMS: first year in operation (AMT/ACP inter-journal SI)". It is not associated with a conference.

*Financial support.* This research was supported by a grant from the National Institute of Environmental Research (NIER), funded by the Korea Ministry of Environment (MOE) of the Republic of Korea (NIER-2023-04-02-050).

*Acknowledgements.* The all authors acknowledge the contribution of the NIER–ESC for providing GEMS data. The GEMS data used in this study were provided by the GEMS Algorithm Team for validation and improvement research purposes.

*Disclaimer.* The English in this document has been checked by professional editors, native speakers of English. For a certificate, please see: (https://drive.google.com/file/d/1phIYsJcsej7eiRsvA30xJixAFum5vLBE/view?usp=sharing)



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
