# Peer review of "Retrieval pseudo BRDF-adjusted surface reflectance at 440nm from Geostationary Environmental Monitoring Spectrometer (GEMS)"

_EGUsphere, 2024_

## Author Comment (AC1)

*Dear. Referee #1*

*We uploaded response letter for the comments of all reviewer and revised manuscript entitled, "Retrieval pseudo BRDF-adjusted surface reflectance at 440 nm from Geostationary Environmental Monitoring Spectrometer (GEMS)".*

*All comments of reviewers were seriously touched by authors and answered in response letter. We tried our responses can satisfy all reviewers, and our manuscript has been more improved by your advice.*

**Comments and Suggestions for Authors as follows (Referee 1):**

1. After reading through the entire paper, I found a few areas that could benefit from further clarification. It would be helpful to expand on the description of the overall process outlined in lines 138-141 to make it easier for readers to follow. Initially, I assumed Figure 1 was a flowchart illustrating the algorithm, with the bottom row representing gap filling. This led me to believe that the method primarily uses BSR while incorporating LER and TOC for gap filling. It took me some time to realize that my interpretation of the flowchart was incorrect. It would be useful if the flowchart clearly indicated the structure of the algorithm, and the text explained that BSR is compared with LER and TOC to validate the BSR results. Even as I write this review, I'm not entirely sure if my understanding of the flowchart is accurate.

➔ Thank you very much for your insightful feedback. We appreciate your observations and agree that the process outlined in lines 138-141 could benefit from further clarification.

In this study, after calculating BSR and LER, we verify the results with the actual TOC data and compare the performance of BSR with LER. If BSR cannot be calculated, we perform gap-filling using LER. However, since this is a minor aspect of the study, it was not initially included in the flowchart. To address potential confusion, we have now provided a more detailed explanation of the study's flow in the text and have revised the flowchart for better clarity.

Below, we have included the revised text and the updated flowchart for your review. The modified parts are as follows: (The previously written parts are in blue and italics, and the newly added/replaced parts are in red and italics.)

*Figure 1 depicts a comprehensive flow chart of the BSR retrieval algorithm, which comprises two primary steps: (1) atmospheric correction and (2) BRDF modeling and BSR retrieval. Subsequently, the constructed GEMS BSR and LER were validated through comparative analysis with the GEMS TOC in this study.The methodology and underlying assumptions are detailed in the subsequent subsections.*

*Figure 1 depicts a comprehensive flow chart of the BSR retrieval algorithm, which comprises two primary steps: (1) atmospheric correction and (2) BRDF modeling and BSR retrieval. To evaluate the applicability of the BSR derived in this study, validations were performed against the GEMS Top-of-Canopy (TOC) data as reference data. Additionally, a comparison was made with the LER data*

*generated using the traditional minimum reflectivity method. Both GEMS BSR and LER were validated against the GEMS TOC data to compare their accuracy, followed by a direct comparison between BSR and LER. The detailed methodology and underlying assumptions are provided in the subsequent subsections.*

[Figure]

**Fig 1. Flow chart of this study (Figure 1 in article; Before the change)**

[Figure]

**Fig 2. Flow chart of this study (Figure 1 in article; After the change)**

2. Additionally, if this method uses LER for gap filling, does it create any discontinuity between the pixels that use BRDF modeling and those where LER is applied? It would be beneficial for the authors to address this and discuss any potential issues with continuity if applicable.

➔ Thank you for your thoughtful comment. We acknowledge that BSR and LER utilize different algorithms, which may result in discontinuities between pixels when applying the gap-filling algorithm. While no obvious discontinuities are visible to the naked eye, it is possible to consider these as discontinuities in a strict sense due to the differing calculation methods.

However, since the number of pixels requiring gap-filling by this method is very small, we believe that introducing additional artificial methods may not be desirable. Instead, we will conduct further work to identify gap-filled areas within the quality flag and other relevant indicators.

3. Regarding the title of Section 4.4, GEMS LER is mentioned as one of the LER databases, but it's unclear whether it was used in Section 4.4. I tried to find a comparison with GEMS LER in this section but couldn't locate any reference to it. If this omission is accurate, it might be better to remove GEMS LER from the section title.

➔ Yes, we have removed "GEMS LER" from the subsection title. Thank you.

4. Lastly, I'm uncertain if it's appropriate to describe this method as a "novel concept" or "novel approach" in the abstract and conclusion. Although this study uses the new hyperspectral sensor GEMS, the methodology (BRDF modeling, LER) itself doesn't seem particularly innovative. Thus, I'm not sure if these terms accurately represent the uniqueness of the approach.

➔ We strongly agree with the comments provided. We initially stated that our study was innovative because it was the first attempt to apply atmospheric correction and BRDF modeling within the official output algorithm of a current environmental satellite such as GEMS. However, as the reviewer mentioned, the methodology itself may not be innovative. In light of this, we have revised the text to convey that it represents the first practical application of this approach. Below are the changes we made to the Abstract and Conclusion. The modified parts are as follows: (The previously written parts are in blue and italics, and the newly added/replaced parts are in red and italics.)

(Line 5-7; In Abstract)
*(Before the change)*
*This study introduces a novel approach to surface reflectance retrieval, termed background surface reflectance (BSR), which leverages a semi-empirical Bidirectional Reflectance Distribution Function (BRDF) model to simulate surface reflectance based on BRDF components*

*(After the change)*
*This study is the first to assess the applicability of background surface reflectance (BSR), derived using a semi-empirical Bidirectional Reflectance Distribution Function (BRDF) model, in an operational environmental satellite algorithm.*

(Line 395-396; In Section 5. Conclusion)

*(Before the change)*

*This study introduced the novel concept of BSR as an alternative output to resolve the output precedence dilemma between land surface reflectance and other L2 outputs applied to GEMS, a hyperspectral satellite observing in the UV-VIS range.*

*(After the change)*

*This study represents the first practical application of BSR as an alternative output to resolve the output precedence dilema between land surface reflectance and other L2 outputs applied to GEMS at 440 nm, evaluating its feasibility for operational use.*
* * *
5. Line 140 : GEMS TOC -> GEMS Top of Canopy (TOC)
* * *
➔ We have revised the relevant sections as suggested. Thank you.
* * *
6. Line 173 : paragraph is duplicated in line 185
* * *
➔ We have reviewed the relevant sections and deleted the redundant parts. Thank you.
* * *
7. Line 212 : does '15 d' means 15 days?
* * *
➔ Yes, that is correct. We have revised the relevant section to 15 days. Thank you.
* * *
8. Line 239 : does 'SFC' meas surface reflectance?
* * *
➔ Yes, SFC stands for Surface Reflectance, which in this study is synonymous with TOC (Top-Of-Canopy) Reflectance. However, for consistency throughout the paper, we have changed it to TOC Reflectance.
* * *
9. Line 373 : I guess author was intending 'GEMS TOC' not 'GEMS, TOC'
* * *
➔ We have revised the relevant sections as suggested. Thank you.

---

## Author Comment (AC2)

**Dear. Referee #2**

We uploaded response letter for the comments of all reviewer and revised manuscript entitled, "Retrieval pseudo BRDF-adjusted surface reflectance at 440 nm from Geostationary Environmental Monitoring Spectrometer (GEMS)".

All comments of reviewers were seriously touched by authors and answered in response letter. We tried our responses can satisfy all reviewers, and our manuscript has been more improved by your advice.

**Comments and Suggestions for Authors as follows (Referee 2):**

1. In the abstract, "This study pioneered the application of the BRDF model to hyperspectral satellite data in the UV-VIS region, aiming to provide more realistic preliminary surface reflectance data". But in the title and the manuscript only 440nm (Page 3, Line 85) is discussed. Then how the authors can demonstrate the application to the boarder spectral range of UV-VIS?

➔ This study is designed to apply the operational algorithm for surface reflectance of GK-2B/GEMS, which observes the UV-VIS wavelength range. It is the first research to apply atmospheric correction and BRDF to an operational algorithm in this spectral range. However, due to limitations in validation data, the study was conducted at 440 nm. The intention behind the description in the manuscript is to explain the potential for future application of this method. Therefore, while this pioneering study confirms the initial feasibility of applying the algorithm, it will need to be extended to the UV wavelength range in the future.

In summary, this research is significant as it is the first to apply RTM-based atmospheric correction and BRDF modeling to derive BSR in the operational surface reflectance algorithm of environmental satellites observing the UV-VIS range. Although the analysis focused on 440 nm, we plan to extend it to the UV range.

However, we understand the point raised by the reviewer and have accordingly revised or removed some potentially overstated sentences in the manuscript as follows. (The previously written parts are in blue and italics, and the newly added/replaced parts are in red and italics.)

(Line 7-9; In Abstract)
*This study pioneered the application of the BRDF model to hyperspectral satellite data at 440 nm  aiming to provide more realistic preliminary surface reflectance data.*

(Line 84-85; In Section 1. Introduction)
*Therefore, in this study, we propose, for the first time, the application of the BRDF model to hyperspectral satellite data  for more realistic preliminary surface reflectance data.*

(Line 395-396; In Section 5. Conclusion)
*(Before the change)*
*This study introduced the novel concept of BSR as an alternative output to resolve the output precedence dilemma between land surface reflectance and other L2 outputs applied to GEMS, a hyperspectral satellite observing in the UV-VIS range.*

*(After the change)*
*This study represents the first practical application of BSR as an alternative output to resolve the output precedence dilema between land surface reflectance and other L2 outputs applied to GEMS at 440 nm, evaluating its feasibility for operational use.*

(Line 425-427; In Section 5. Conclusion)
*Although limitations exist, such as the challenge of capturing sudden changes in surface characteristics such as snow or ice cover, our research is pioneering in its application to BRDF modeling and evaluation in hyperspectal*  *observation satellite studies.*
* * *
> 2. Page 3, Line 80 "This approach reflects both the high temporal resolution of GLER and the advantages of DLER's own BRDF". Can you explain more what advantage of DLER is preserved?

➔ As stated in the text, GLER data is a data reproduced through inversion of LER based on MODIS BRDF data, and has the advantage of having a higher temporal resolution compared to existing LER data while considering a high BRDF. However, the wavelength produced by MODIS and the wavelength produced by GLER do not match, but because the reflectance ratio of 440 / 466 nm is used to calculate GLER at 440 nm, there is a concern that the BRDF effect is assumed to be linear with reflectivity ratio.

  DLER data, similar to existing LER data, is data reproduced based on reflectivity accumulated over several years, but what differs from LER is that it takes into account the BRDF effect according to the satellite's viewing angle (VZA). Unlike GLER, which is produced based on MODIS BRDF data, DLER has the advantage of being able to individually consider BRDF effects at all calculated wavelengths. (As written in the text, TROPOMI DLER can be produced by calculating empirical coefficients for 9 viewing angle ranges and applying them.) However, like the existing traditional LER method, this is climate data that is not calculated in real time.

  Therefore, the BSR proposed in this study is a product that contains both the advantages of GLER's high temporal resolution and DLER's own BRDF consideration at the wavelength to be calculated.
  As you pointed out, the explanation seems insufficient. Accordingly, the advantages of DLER were explained in more detail. The modified parts are as follows: (The previously written parts are in blue and italics, and the newly added/replaced parts are in red and italics.)

(Line 70-78; Section 1. Introduction)

*In addition, the Satellite Application Facility on Atmospheric Composition Monitoring (AC SAF) and the European Space Agency (ESA) provided a DLER database that considers the viewing geometry from GOME-2 (Tilstra et al., 2021) and the Tropospheric Monitoring Instrument (TROPOMI) (Tilstra et al., 2023). DLER data, similar to existing LER data, is reproduced based on reflectivity accumulated over several years. What differs from LER is that it takes into account the BRDF effect according to the satellite's viewing angles. The DLER database introduced by Tilstra et al. (2021) considers anisotropic features at various viewing angles as an advantage of polar and sun-synchronous orbit satellites. For example, GOME-2 and TROPOMI DLER calculations were performed using the regression coefficients calculated for the 5 and 9 view containers, respectively, allowing them to simulate BRDF effects influenced by the satellite's viewing angles.Therefore, unlike GLER may overlook the BRDF effects that change with wavelengths, DLER can simulate BRDF effects for all individual wavelengths. However, these data are constructed from climatology, such as the LER database, and are not updated annually but provided only once a month, making it difficult to reflect the changing land surface characteristics in real time. Additionally, there are limitations in reflecting the characteristics of indicators that change frequently with a single fixed coefficient, and it is difficult to consider the influence of other geometric conditions (such as the solar zenith angle).*
* * *
3. Page 4, Line 113, "In this study, we utilized 550 nm AOD data to perform atmospheric correction." Why choose 550nm to conduct atmospheric correction if 440nm is the focus of this study?
* * *
➔ Most satellite AOD algorithms calculate the 550 nm AOD because the 550 nm wavelength is important because it is most scattered in the atmosphere and is widely used in various chemical models. Therefore, the general reference wavelength for most satellite AOD products is set to 550 nm. The 6SV RTM used to perform atmospheric correction in this study is also structured to input the AOD value at 550 nm. Therefore, the AOD at 550 nm was used as input in this study.

We acknowledged that this explanation was insufficient and additionally mentioned in the text the reason for using the AOD of 550 nm as follows. (The previously written parts are in blue and italics, and the newly added/replaced parts are in red and italics.)

(Line 110-114; In Section 2.2. Geostationary Environment Monitoring Spectrometer)

*The GEMS AERAOD data provided AOD values for three wavelengths (354, 443, and 550 nm). The GEMS AOD at 443 nm shows high accuracy with a strong positive correlation coefficient (R-value) of about 0.89 and a low root mean squared error (RMSE) of 0.15 after validation with the AERONET observed AOD (Cho et al., 2023). In this study, we utilized 550 nm AOD data to perform atmospheric correction. Most satellite AOD algorithms calculate the 550 nm AOD due to its significant scattering in the atmosphere and its widespread use in various chemical models. Additionally, the 6SV RTM used for atmospheric correction in this study is designed to input the AOD value at 550 nm.*
* * *
4. Page 4, Line 115, why Pandora is better?
* * *
➔ This does not mean that Pandora data is better than satellite data (such as GEMS, TROPOMI and OMPS), and when the TCO (Total Column Ozone) of GEMS, TROPOMI, and OMPS were verified

based on Pandora, the accuracy of the GEMS TCO output was similar to or slightly better than that of OMPS and TROPOMI. This was mentioned to confirm the stability of GEMS TCO data.
* * *
5. Page 6, Line 119, how the aerosol information is determined for the atmospheric transmittance? Is it fixed?

6. Page 6, Line 119, there are several transmittance variables used here, are they all defined in the same way or not? Namely whether they are for radiance or irradiance?

7. Page 6, Line 119, How the spherical albedo is defined?
* * *
→ We think it will be easier for you to understand the answers to the three questions above if we explain them all at once. Atmospheric correction in this study can be performed based on the xap, xb, and xc coefficients calculated through 6SV RTM (Equation 1~4 in the text). To do this, input of angle components, aerosols, and atmospheric components is required, as shown in Table 1.

Therefore, in this study, the three angular components SZA, VZA, and RAA, TCO, AOD, and Terrain height were selected as atmospheric correction input variables, and the aerosol model was set to Contienetal. However, RTM-based calculations have slow calculation speeds, which limits their application to real-time operational algorithms. Therefore, we calculated the coefficients for various conditions (min, max, increment in Table 1) in advance and saved them in LUT format before using them.

So the answer to question 5 is: The Continental aerosol model in 6SV was used, and a total of 11 AODs (0.01, 0.05, 0.1, 0.15, 0.2, 0.3, 0.4, 0.6, 0.8, 1.0, 1.5) were simulated as shown in Table 1.

Additionally, atomspheric transmittance, several transmittances, spherical albedo, etc. mentioned in questions 6 and 7 are all values that can be obtained as output when input variables are entered into 6SV RTM and simulated. These calculation formulas are explained in detail in the 6SV Manual (Part1), and the formulas below are quoted from 6SV Manual Part1 (Vermote et al., 2006).

**The description of the variables in the formula is as follows:**

1. $S(\lambda)$: Spectral filter function, representing the filter response at a specific wavelength $\lambda$

2. $E_s$: Solar spectral irradiance, representing the solar radiation energy at a specific wavelength $\lambda$

3. $\mu_s$ : Cosine of the solar zenith angle

4. $T_{gi}(\theta)$: Gas transmittance for the i-th gas, $T_g(\lambda, \theta)$ represents the transmittance by gases at $\lambda$ and $\theta$

5. $\tau(\theta)$: Optical depth, representing the optical thickness of the atmosphere or aerosols at $\lambda$ and $\theta$

6. $T_r(\theta_s)$: Rayleigh transmittance at the solar zenith angle

7. $T_a(\theta_s)$: Aerosol transmittance at the solar zenith angle

8. $S^R$ : Spherical albedo for Rayleigh scattering

9. $S^A$: Spherical albedo for aerosols

10. $S^T$: Total spherical albedo

6S User Guide Version 3, November 2006

| Gas Trans: $H_2O$, $O_3$, $CO_2$, $O_2$ $N_2O$, $CH_4$, CO | DOWNWARD | UPWARD | TOTAL |
|---|---|---|---|
|  | $\dfrac{\int S(\lambda)\mu_s E_\lambda \prod_{i=1}^{7} Tg_i^\lambda(\theta_s)d\lambda}{\int S(\lambda)\mu_s E_\lambda d\lambda}$ | $\dfrac{\int S(\lambda)\mu_s E_\lambda \prod_{i=1}^{7} Tg_i^\lambda(\theta_v)d\lambda}{\int S(\lambda)\mu_s E_\lambda d\lambda}$ | $\dfrac{\int S(\lambda)\mu_s E_\lambda \prod_{i=1}^{7} Tg_i^\lambda(\theta_s,\theta_v)d\lambda}{\int S(\lambda)\mu_s E_\lambda d\lambda}$ |
| Rayleigh transmittance. Aerosols transmittance. Total transmittance. | $\dfrac{\int S(\lambda)\mu_s E_\lambda T_\lambda(\theta_s)\,d\lambda}{\int S(\lambda)\mu_s E_\lambda d\lambda}$ | $\dfrac{\int S(\lambda)\mu_s E_\lambda T_\lambda(\theta_v)\,d\lambda}{\int S(\lambda)\mu_s E_\lambda d\lambda}$ | $\dfrac{\int S(\lambda)\mu_s E_\lambda T_\lambda(\theta_s)T_\lambda(\theta_v)\,d\lambda}{\int S(\lambda)\mu_s E_\lambda \,d\lambda}$ |

|  | Rayleigh | Aerosols | Total |
|---|---|---|---|
| Spherical albedo | $\dfrac{\int S(\lambda)\mu_s E_\lambda S^R d\lambda}{\int S(\lambda)\mu_s E_\lambda d\lambda}$ | $S^A$ | $S^T$ |

**Fig 1. Description of the outputs in 6SV (6SV Manual Part 1; Vermote et al., 2006)**
* * *
8. Page 7, Table 1, continental aerosol, how it is defined and determined? Do you use an optimization procedure to retrieve the aerosol properties, such as AOD? Do you assume a surface model when determine the aerosol properties?
* * *
➔ The Continental model utilized in this study is an aerosol model predefined within 6SV. This is a model produced by mixing three basic components: dust-like component (DUST), water-soluble component (WATE), and soot component (SOOT). The weighted averages of DUST, WATE, and SOOT in that order are 0.7 and 0.29, 0.01, respectively. Additionally, when simulating through 6SV, the land cover type was set to vegetation.

Additionally, in this study, we introduced the concept of BSR to address the retrieval paradox between surface reflectance and AOD mentioned in the introduction. Therefore, the algorithm used is not intended to independently retrieve AOD. Instead, GEMS AOD was utilized to calculate TOC as input data for simulating BSR. As a result, no additional optimization procedures were performed for AOD in this study.
* * *
9. Page 8, Line 195-202, the whole paragraph is duplicated from above!
* * *
➔ Thank you. We checked and deleted the duplicate parts.
* * *
10. Page 8, Equation 5, is there spectral dependency of those parameters?
* * *
➔ Certainly. The BRDF kernels (f1, f2; In Equation 5) used in this study were all based on the Roujean model (Roujean et al., 1992). The parameters K0, K1, and K2 in Equation 5 can be empirically derived from these f1 and f2 kernels. According to the study by Roujean et al. (1992), these parameters vary depending on the Earth's surface and the observed wavelength. For example, the K2 parameter can be about 3 to 5 times larger in the near-infrared than in the visible range.

Therefore, while the values at a specific point in the same channel remain almost unchanged, reflectance values vary with different channels, indicating that these variables change with the spectrum.

**[Reference #1]**

Roujean, J. L., Leroy, M., & Deschamps, P. Y. (1992). A bidirectional reflectance model of the Earth's surface for the correction of remote sensing data. Journal of Geophysical Research: Atmospheres, 97(D18), 20455-20468.
* * *
11. Page 8, Line 207, how time frame matters here?
* * *
➜ In BRDF modeling, the temporal variable is extremely important and can be described as the compositing period in BRDF modeling. As the compositing period lengthens, the number of available pixels increases, leading to a higher sample size for BRDF modeling and thus achieving more accurate BRDF modeling. However, a longer compositing period may overlook the reflectance distribution characteristics due to real-time changes in the surface. Therefore, the compositing period for BRDF is selected by considering the number of samples available based on the satellite's revisit cycle and the resulting accuracy of BRDF modeling. In this study, the compositing period was empirically set to 15 days.
* * *
12. Page 10, Line 231, what is the age variable? Where is it in the equation?
* * *
➜ To address the gaps that may occur due to insufficient observations during BRDF modeling, we implemented an additional design where BRDF parameters from up to 5 days prior are used when current parameters are not available for the same pixel. We defined and utilized an "Age" variable, which indicates how many days old the BRDF parameters used for gap-filling are. For instance, if BRDF parameters from 3 days ago are used, the Age variable is assigned a value of 3. If the next day also lacks newly calculated BRDF parameters and uses the same date's BRDF parameters, the Age variable is assigned a value of 4. Conversely, if new BRDF parameters are calculated, the Age variable is reset to 0. When the Age variable reaches 5, the parameters are no longer used.

We acknowledged that this explanation was insufficient and additionally mentioned in the text the reason for using the "Age" variable as follows. (The previously written parts are in blue and italics, and the newly added/replaced parts are in red and italics.)

(Line 217-221; In Section 3.2. BSR retrieval through BRDF modeling)

*Insufficient observations lead to uncalculated BRDF parameters, resulting in missing BSR values, which are crucial for other L2 algorithms. To address this, a gap-filling process was implemented, utilizing the "Age" variable in BRDF modeling. This method uses previously calculated BRDF parameters up to 5 days of age to fill gaps when current parameters are unavailable. In this process, the Age variable was defined and utilized, indicating how many days old the BRDF parameters used for gap-filling are. For example, if BRDF parameters from 3 days ago are used, the Age variable is*

*assigned a value of 3. If the next day also BRDF parameters are not be calculated, and uses the same date's BRDF parameters, the Age variable is assigned a value of 4. Conversely, if new BRDF parameters are calculated, the Age variable is reset to 0. If the Age variable exceeds 5, the parameters are no longer used. To summarize,* the Age variable indicates the number of days since the last valid BRDF calculation, reset to 0 after 5 days, or upon the BRDF parameter is calculated.
* * *
13. Page 10, Line 246, "The GK-2A AMI albedo output establishes the standard for good quality BSR, requiring seven or more observations and a BRDF RMSE of 0.07 or lower" How the number of observations impact accuracy? Does this relate to the coverage of the angular range of the BRDF? Is the geometry limited for a geostationary satellite?
* * *
➔ The accuracy of BRDF modeling is influenced by the number of observations, errors in surface reflectance (TOC), and the sampling conditions of the sun-satellite angles (such as SZA and VZA). When the number of observations is low, errors in surface reflectance can be mistakenly simulated as BRDF effects. Therefore, a sufficient number of observations is essential for error reduction and stable BRDF effect simulation.

Additionally, as you mentioned, geostationary satellites like GEMS, unlike polar-orbiting satellites, exhibit significant variability in observed solar zenith angles while having minimal changes in satellite (viewing) zenith angles. This can lead to greater errors in BRDF inversion studies for conditions that are not sampled. However, this study performs BRDF modeling based on the angle components sampled by the GEMS satellite, and therefore does not consider the accuracy of BRDF at unsampled angles.

Therefore, we extracted BSR good quality pixels based on the quality flag (number of observations) applied to the AMI satellite, which observes similar angles to the geostationary GEMS satellite, and the BRDF good quality criteria (BRDF RMSE) provided by the MODIS satellite.
* * *
14. Page 11, Figure 3 shows good agreement. How does the comparison relate to solar geometry? Do you have examples what the BRDF looks like?
* * *
➔ The BRDF models used in this study are based on the Roujean BRDF model (Roujean et al., 1992), and the formulas for the two kernels (f1, f2) are as follows. Consequently, as the solar or viewing geometry changes, the reflectance changes according to the formulas below. Additionally, the two kernel values of the Roujean BRDF model show the distribution of values according to SZA and RAA (VZA = 50), as illustrated in Figures 2 and 3 below.

$$f_1(\theta_s, \theta_v, \phi) = \frac{1}{2\pi}[(\pi - \phi)\cos\phi + \sin\phi]\tan\theta_s \tan\theta_v$$
$$- \frac{1}{\pi}\left(\tan\theta_v + \tan\theta_s + \sqrt{\tan^2\theta_v + \tan^2\theta_s - 2\tan\theta_s\tan\theta_v\cos\phi}\right) \tag{1}$$

$$f_2(\theta_s, \theta_v, \phi) = \frac{4}{3\pi} \cdot \frac{1}{\cos\theta_s + \cos\theta_v}\left[\left(\frac{\pi}{2} - \zeta\right)\cos\zeta + \sin\zeta\right] - \frac{1}{3} \tag{2}$$

$$\zeta = \arccos[\cos\theta_v \cos\theta_s + \sin\theta_v \sin\theta_s \cos\phi] \tag{3}$$

[Figure]

**Fig 2. Roujean BRDF geometric kernel (f1) variation with SZA and RAA (VZA = 50)**

[Figure]

**Fig 3. Roujean BRDF volumetric kernel (f2) variation with SZA and RAA (VZA = 50)**

15. Page 13, Figure 4, what is the solar and view geometry here?

➔ TOC and BSR utilized solar and viewing geometry at the time of actual observation (L1C data), and LER was calculated using the minimum reflectance method and did not consider geometry separately.

16. Page 13, Line 296, how you separate the four land types? Could you show them in a map?

➔ The analysis was conducted based on MODIS land cover data. The data used is from the MODIS/Terra+Aqua Land Cover Type Yearly L3 Global 0.05Deg CMG V061 (MCD12C1), specifically from the year 2021. Figure 2 below represents the land cover according to the IGBP

classification within the MODIS Land Cover data. The values for the four land cover types used in this study (Grassland, Cropland, Shrubland and Urban) are displayed in Figure 3. The average values for all four land cover types were utilized.

[Figure]

**Fig 4. MODIS Landcover (IGBP classification system; 2021)**

[Figure]

**Fig 5. The area within the MODIS land cover used in this study (Grassland, Cropland, Shurbland, Urban)**

17. Page 15, Line 312, how AOD is determined??

➔ The section (4.2.2.) does not involve calculating the actual AOD and performing analysis, but rather, the analysis is based on the changes in AOD according to the variations in blue channel surface reflectance presented in the study by Li et al., (2012). Using TOC reflectance as a reference value, we utilized BSR and LER as alternative reflectances. We then evaluated the potential errors in AOD estimation by creating a regression equation based on the changes presented in Table 2 of the Li et al. study.

Please refer to the paper mentioned below for further details.

**[Reference #2]**
Li, S., Chen, L., Tao, J., Han, D., Wang, Z., Su, L., ... & Yu, C. (2012). Retrieval of aerosol optical depth over bright targets in the urban areas of North China during winter. Science China Earth Sciences, 55, 1545-1553.
* * *
18. Page 16, Line 398, how the surface reflectance is observed from ground site? What is the spatial coverage? Any particular viewing geometry the reflectance is defined?
* * *
➔ In this study, surface reflectance at the Baotou Sandy Site (BSCN) of RadCalNet, a desert area with minimal reflectance variability. Surface reflectance at the Baotou ground site is measured using an automated observation system developed by the Academy of Opto-Electronics (AOE) at the Chinese Academy of Sciences (CAS). The system includes CR-250 spectrometers, which cover a spectral range of 380 nm to 1080 nm with a 2 nm resolution and a 3-degree field of view.

The spatial coverage for the sandy site at Baotou is $300 \times 300$ meters (center coordinates: 40.8658 N, 109.6155 E), with a spatial heterogeneity of less than 2.5% from 618 nm to 868 nm. Observations are made every 2 minutes from 08:30 to 17:30 local time, and the data is transferred to the data center in Beijing via the internet.
* * *
19. Page 17, Figure 7, it seems the surface reflectance variability is quite narrow comparing with the one derived from satellite. Is this because the ground site covers a small spatial range, while satellite measures a larger range?
* * *
➔ Yes, that's correct. Although the RadCalNet BSCN site is a desert area with relatively consistent reflectance distribution, its spatial coverage is only 0.3 x 0.3 km, which is significantly smaller compared to satellite observations (GEMS: 7 x 8 km at Seoul, TROPOMI: 0.125 x 0.125 degrees, OMI: 7 x 7 km at nadir). Consequently, the reflectance characteristics from areas around the site, which are not desert, are also included, leading to greater variability in the values compared to those observed at the BSCN site. Please refer to sections 2.3 and 2.4 of the QA4EO-WGCV-RadCalNet-BTCN-Q-v3.pdf linked below for photos of the BSCN site and its surroundings.

In addition, data from the time zone closest to the ground observation was used for verification. However, differences may occur due to the sun-target and target-sensor geometry characteristics inherent in satellite observation.

( URL : https://www.radcalnet.org/#!/sites/BTCN )
* * *
20. Page 20, line 396, "This study introduced the novel concept of BSR as an alternative output to resolve the output precedence dilemma between land surface reflectance and other L2 outputs applied to GEMS, a hyperspectral satellite observing in the UV-VIS range. " Again, this study only discuss one wavelength, so the conclusion to the application to UV-VIS may be not complete.

➔ The response to this question is the same as for question 1; therefore, a separate answer was not included. We greatly appreciate the reviewer's comments and have made the necessary revisions. The updated section is included below. (The previously written parts are in blue and italics, and the newly added/replaced parts are in red and italics)

(Line 395-396; In Section 5. Conclusion)
*(Before the change)*
*This study introduced the novel concept of BSR as an alternative output to resolve the output precedence dilemma between land surface reflectance and other L2 outputs applied to GEMS, a hyperspectral satellite observing in the UV-VIS range.*

*(After the change)*
*This study represents the first practical application of BSR as an alternative output to resolve the output precedence dilema between land surface reflectance and other L2 outputs applied to GEMS at 440 nm, evaluating its feasibility for operational use.*

> 21. Page 21, line 404, "The simulation performance of the GEMS BSR was 3% more accurate than that of the GEMS LER data in terms of the rRMSE over the entire study period based on TOC". Is 3% significant? How does this compare with the measurement uncertainty from GEMS?

➔ As mentioned in the introduction, the underestimation of surface reflectivity that occurs when LER based on the minimum reflectivity method is used as an alternative reflectivity directly leads to the overestimation of AOD and affects other L2 products such as clouds and gas products (NO2, SO2).

As noted in section 4.2.2, "Surface Reflectance Influence on AOD Variability in Cropland and Urban Areas," while the numerical improvement in overall accuracy is approximately 3%, the influence on AOD shows that BSR is up to about 10% better than LER, particularly in downtown areas during winter.

Therefore, the improvement of BSR is meaningful in this respect as it can address the inherent problems of LER.

---

## Referee Report (RR1)

The manuscript titled by "Retrieval pseudo BRDF-adjusted surface reflectance at 440 nm from Geostationary Environmental Monitoring Spectometer (GEMS)" shows detailed explanation of GEMS surface reflectance retrieval algorithm. The manuscript describes well for the purpose of algorithm, application, and its validation results. However, some correction points have remained, such as confused word using or lack of detailed figure captions. Therefore, this manuscript will be revised before accepting the journal.

Detailed comments are listed below

1) Some words are rarely used in the algorithm fields. The algorithm is just estimation and guessed values from several assumptions. Therefore, 'calculation' is rarely used. In addition, 'outputs' in section 2.2 is also not frequently used for the retrieval products, and 'generation' in section 3.3 is also rarely used. Please check the word related to the retrieval and retrieved dataset explanations and correct it.

2) P1 L19: land surface reflectance is essential to the satellite remote sensing, but the surface signal is negligible to the 'ground' remote sensing. Therefore, please change the 'remote-sensing' to 'satellite remote sensing'. Also, many points in the manuscript have similar word using. Please correct it.

3) P2 L32-L41: From this manuscript, all the satellite algorithms are based on the minimum reflectance technique. However, it is doubtful that the minimum reflectance technique is one of the method for the surface reflectance retrieval. In addition, the GOME and OMI climatological surface reflectance data is partly different to identify the maximized surface signals. The detailed previous retrieval algorithms and other reflectance identification techniques are required.

4) P2 L42-58: For the Level 2 scientific product explanation, the author needs to include the references, such as ATBDs or papers of AMF errors.

5) P3 L60: References will be added with respective to the different products of LER.

6) P3 L84-91: The author shows the suggestion of the methods. However, the purpose of this study and importance of study are not included in the Introduction. Please include the purpose of study and also describe the sections.

7) P4: Please change the order of section. Section 2.1 will be shown after the Section 2.2.

8) P4 L104: Level-1C → Level 1C (L1C)

9) P4 L110: 'it does not provide an official cloud mask': The GEMS officially provides the cloud product (GEMS CLD). It is confusing to the reader. What is this sentence means?

10) P4 L110-111: For the clear-sky identification, is this study uses the CCP > 1000 hPa,

addition to the ECF<0.2 and CCP = 1013hPa? It has advantage of clear-sky identification from falsely detected as cloudy pixels. However, how about the cloud conditions in real, but cloudy from GEMS CLD?

11) Change "R-value" to 'r'

12) P4 L116-117: The author shows the accuracy of GEMS AOD at 443 nm. However, I am doubtful that this accuracy results is not guaranteed different spectral AOD accuracy. During the spectral conversion of AOD, the error will be enhanced. Why don't the author use the 443 nm AOD values?

13) P5 L125-130: For the supplement of GEMS AOD gap, this study used the CAMS AOD values. However, the CAMS AOD have significant biases as compared to the GEMS AOD. The AOD bias between GEMS and CAMS will be affected to the discontinuous spatial distribution of AOD, and thus affecting to the surface reflectance calculation. How much this bias affecting the surface reflectance estimation? In addition, how did this study correct the AOD bias between CAMS and GEMS?

14) Section 3: Before beginning the section 3, the author needs to the definition and calculation process of Top-of-Canopy from GEMS.

15) P5 L147: what is the 'traditional minimum reflectivity method'? Please add the details and references.

16) Figure 1: Change the caption to "Flowchart of GEMS BSR algorithm."

17) Equation 1-5 and related sentences: For the equation writing, please use the subscript. All the equation variables are not use the subscript and it may confusing to the equation. In addition, all the equation variables will be clarify in the manuscript.

18) P6 L160: For the 6SV RTM, How to adopting the spectral response function of GEMS?

19) Table 1: For the Aerosol type, what is "continental"? Do you have any detailed aerosol optical and physical properties, or related references? In addition, the TCO value range is too narrow. In East Asia, the total ozone is ranged from 200-600 DU based on the daily data. 250-350 DU ranges are too narrow. Do this narrow TCO range affect the surface reflectance retrieval?

20) P9 L231: This study uses the 15-day period for the clear-sky identification. However, 15-day is too narrow temporal window. From the below reference, the 30-day temporal window is essential to identify the clear-sky conditions.
Park, S. S., Yu, J. E., Lim, H., and Lee, Y. G.: Temporal variation of surface reflectance and cloud fraction used to identify background aerosol retrieval information over East Asia, Atmos. Environ., 309, 119916.

21) Figure 3: The intercomparison has been done by using the GEMS TOC. By the comparison with GEMS TOC, how to be explain the significance of accuracy improvements?

22) P11 L275: GEMS SFC is not defined in this manuscript.

23) P13 L299: Please add the references and detailed products used in this study.

24) P15 L315: From Table 1, the AOD bin range is not exceeded to 1.5. How to be analyzed the AOD ranges up to 2.0?

25) Figure 8: Is it possible to include the direct comparison between GEMS BSR and LERs?

---

## Author Response (AR2)

*Dear. Referee #3*

*We uploaded response letter for the comments of all reviewer and revised manuscript entitled, "Retrieval pseudo BRDF-adjusted surface reflectance at 440 nm from Geostationary Environmental Monitoring Spectrometer (GEMS)".*

*All comments of reviewers were seriously touched by authors and answered in response letter. We tried our responses can satisfy all reviewers, and our manuscript has been more improved by your advice.*

*The line numbers written in this response are based on the Author's tracked changed file.*

**Comments and Suggestions for Authors as follows (Referee 3):**

> 1. Some words are rarely used in the algorithm fields. The algorithm is just estimation and guessed values from several assumptions. Therefore, 'calculation' is rarely used. In addition, 'outputs' in section 2.2 is also not frequently used for the retrieval products, and 'generation' in section 3.3 is also rarely used. Please check the word related to the retrieval and retrieved dataset explanations and correct it.

➔ Thank you very much for your insightful feedback. I largely agree with your observation that certain words are rarely used in the context of algorithm fields. Specifically, terms like "generation" and "output" are indeed infrequently used when compared to other studies. However, in research papers related to surface reflectance and albedo, the term "calculation" is commonly employed. Consequently, I have made changes to all instances of "generation" and "output" throughout the text. I have also revised some instances of "calculation" where it could potentially cause confusion.

All instances of the word "output" in the text have been changed to "product".

Below is the attached text with the revised parts other than the relevant parts. The modified parts are as follows: (The previously written parts are in blue and italics, deleted parts are blue and , and the newly added/replaced parts are in red and italics.)

*(Line 25-28; In Section 1. Introduction)*
*In the retrieval  of surface reflectance, aerosol optical depth (AOD) and atmospheric gas products (such as ozone, total precipitable water (TPW), nitrogen dioxide (NO2), etc.) are fundamental parameters. Simultaneously, surface reflectance is crucial for the retrieval  of these atmospheric constituents and forms an essential component of their product .*

*(Line 255; Change the section title)*
*(Before the change)*
*3.3. GEMS LER generation*

*(After the change)*
*3.3. GEMS LER assumption*

2. P1 L19: land surface reflectance is essential to the satellite remote sensing, but the surface signal is negligible to the 'ground' remote sensing. Therefore, please change the 'remote-sensing' to 'satellite remote sensing'. Also, many points in the manuscript have similar word using. Please correct it

→ Thank you very much for your insightful feedback. All parts related to the comment have been edited.

Below, we have included the revised text for your review. The modified parts are as follows: (The previously written parts are in blue and italics, and the newly added/replaced parts are in red and italics.)

*(In Abstract)*
*In satellite remote sensing applications, enhancing the precision of level 2 (L2) algorithms relies heavily on the accurate estimation of the surface reflectance across the ultraviolet (UV) to visible (VIS) spectrum.*

*(In Abstract)*
*These findings present a promising avenue for enhancing the accuracy of surface reflectance retrieval from hyperspectral satellite data, thereby advancing the practical applications of satellite remote sensing algorithms.*

*(Line 17-19; In Section 1. Introduction)*
*Because surface reflectance is utilized in remote- sensing systems to derive various geophysical, chemical, and biological variables (Veefkind et al., 2006; Noguchi et al., 2014), accurate satellite observations of land surface reflectance are essential for developing accurate satellite remote-sensing algorithms.*

*(Line 180-181; In Section 3.1. Atmospheric correction)*
*Atmospheric correction plays a pivotal role in satellite remote sensing by rectifying distortions caused by atmospheric effects, which can vary based on different geometries and atmospheric conditions.*

*(Line 221-222; In Section 3.2. BSR retrieval through BRDF modeling)*
*Consequently, semi-empirical BRDF models are widely employed in satellite remote sensing.*

3. P2 L32-L41: From this manuscript, all the satellite algorithms are based on the minimum reflectance technique. However, it is doubtful that the minimum reflectance technique is one of the method for the surface reflectance retrieval. In addition, the GOME and OMI climatological surface reflectance data is partly different to identify the maximized surface signals. The detailed previous retrieval algorithms and other reflectance identification techniques are required.

→ The LER (Lambertian Equivalent Reflectivity) algorithm we are most familiar with is a surface reflectance calculated based on the minimum reflectance technique. The minimum reflectance technique, used by many satellite algorithms, identifies the lowest reflectance value observed over a given period of time, assumed to represent clear sky conditions. The method is simple and often effective in areas with low reflectance variability, such as over the ocean or over stable land surfaces.

However, this method has limitations and is not universally accepted as the best method for all surface types. For example, in areas with high surface reflectance variability, such as deserts, snow-covered

areas, or vegetated areas, the minimum value may not accurately represent typical surface conditions, and in areas where occasional low reflectance occurs due to temporary clouds or shadows, the reflectance may not reflect the true surface properties.

Therefore, the GOME-2 LER product generates an output called "MODE-LER", which is available in addition to the more familiar "MIN-LER" method of selecting the minimum reflectance. Instead of selecting the minimum value that occurred during the synthesis period, the MODE-LER method adopts the most frequently occurring reflectance. This can provide a more representative measure of typical surface reflectance, especially in areas where surface conditions are highly variable.

This approach suggests that applying the minimum reflectance technique to calculate surface reflectance is limited in its ability to account for the anisotropic reflectance properties of the ground surface. However, the MODE-LER method is also a climatological dataset and is limited in its ability to parameterize all of the anisotropic reflectance properties of the ground surface, which are changing in real time. Therefore, in this study, a more realistic surface reflectance simulation was performed based on the BRDF model.

However, as the reviewer mentioned, there are several methods to calculate the reflectance of the land surface other than the minimum reflectance method in the OMI and GOME-2 satellite in situ algorithms, so we have added a reference to this in the text.

Below, we have have attached the added text for your review. The modified parts are as follows: (The previously written parts are in blue and italics, and the newly added/replaced parts are in red and italics.)

*(Line 32-47; In Section 1. Introduction)*

*Most satellite algorithms that focus on observing the UV-VIS region, such as the Total Ozone Mapping Spectrometer (TOMS) (Herman and Celarier, 1997), Global Ozone Monitoring Experiment (GOME) (Koelemeijer et al., 2003), and Ozone Monitoring Instrument (OMI) (Kleipool et al., 2008) produce a priori surface reflectance products called Lambertian equivalent reflectance (LER). The LER archive is a climatology database calculated using the minimum reflectance method under the assumption of a Lambertian surface. The minimum reflectance technique uses the lowest observed ground reflectance for the same pixel within the compositing period. This technique assumes that the minimum value of the surface reflectance generated during the synthesis period minimizes the effects of the atmosphere and clouds, and adopts it as a stable value in a clear sky. This method has the advantage of being simple to implement, but has a limitation in that it cannot consider realistic surface reflection properties and can easily underestimate the actual surface reflectance. Occasionally, overcalculations can occur because of a failure to reflect the characteristics of changes in the indicators in real time. Therefore, to idetify more realistic surface signal, the GOME-2 (Tilstra et al., 2017) LER products have introduced the "MODE-LER" method alongside the "MIN-LER" approach. While the MIN-LER method selects the minimum reflectance observed during the synthesis period as well known, the MODE-LER method identifies the most frequently occurring reflectance value. This approach provides a more representative measure of typical surface reflectance, especially in regions with highly variable surface conditions. Although the MODE-LER method offers improvements rather than minimum reflectacne method, it remains a climatological dataset and cannot fully parameterize the anisotropic reflectance properties of surfaces, which change in real time.*

> 4. P2 L42-58: For the Level 2 scientific product explanation, the author needs to include the references, such as ATBDs or papers of AMF errors.

➔ We recognized that there was a lack of references, and we appreciate your assistance. An additional paper was cited that emphasized the importance of considering the anisotropic nature of surface reflectivity in each Level 2 output algorithm (such as AOD, HCHO, NO2 and SO2). Additionally, to highlight the significance of the aspects explained directly in the paper, previously indirect references were changed to direct mentions.

The corrections are listed below: The modified parts are as follows: (The previously written parts are in blue and italics, and the newly added/replaced parts are in red and italics.)

*(In Line 48-50; In Section 1. Introduction)*
*Under- and over-estimated surface reflectance arising from neglecting surface anisotropy can significantly compromise the accuracy of other satellite-derived products such as AOD (**Kaufman et al., 1997**), formaldehyde (HCHO) (**De Smedt et al., 2018; Howlett et al., 2023**), NO2 and SO2 (**Leitão et al., 2010; McLinden et al., 2014**).*

*(In Line 50-53; In Section 1. Introduction)*
*For instance, an underestimated surface reflectance may result in an overestimation of the AOD (**Mei et al., 2014**), affecting the accuracy of aerosol concentration estimate. Conversely, if the surface reflectance is overestimated, the opposite effects would occur (**Chen et al., 2021; Wang et al.,2019**).*

*(In Line 53-61; In Section 1. Introduction)*
*(Before the change)*
*Based on prior studies, the variation of the surface reflectance 0.05 in the blue channel can lead underestimation of approximately -0.17 in the range where the AOD is less than 0.4 (Li et al., 2012). Additionally, an error of 0.01 in the surface reflectance in the UV region results in an AOD error of 0.1 (Veefkinf et al., 2000). In addition, errors in cloud retrieval can also occur, affecting cloud properties such as cloud fraction and optical thickness.*

*(After the change)*
*Li et al. (2012) found that a variation of 0.05 in the surface reflectance in the blue channel can lead to an underestimation of approximately -0.17 in the range where the AOD is less than 0.4. Veefkind et al. (2000) indicated that an error of 0.01 in the surface reflectance in the UV region results in an AOD error of 0.1. Furthermore, **Platnick et al. (2001); Letu et al. (2020)** observed that errors in cloud retrieval can also occur, affecting cloud properties such as cloud fraction and optical thickness*

**[References #1]**
**Kaufman**, Y. J., Wald, A. E., Remer, L. A., Gao, B. C., Li, R. R., & Flynn, L. (**1997**). The MODIS 2.1-/spl mu/m channel-correlation with visible reflectance for use in remote sensing of aerosol. IEEE transactions on Geoscience and Remote Sensing, 35(5), 1286-1298.

**[References #2]**
**De Smedt**, I., Theys, N., Yu, H., Danckaert, T., Lerot, C., Compernolle, S., ... & Veefkind, P. (**2018**). Algorithm theoretical baseline for formaldehyde retrievals from S5P TROPOMI and from the QA4ECV project. Atmospheric Measurement Techniques, 11(4), 2395-2426.

**[References #3]**
**Howlett**, C., González Abad, G., Chan Miller, C., Nowlan, C. R., Ayazpour, Z., & Zhu, L. (**2023**). The influence of snow cover on Ozone Monitor Instrument formaldehyde observations. Atmósfera, 37.

**[References #4]**
**Leitão**, J., Richter, A., Vrekoussis, M., Kokhanovsky, A., Zhang, Q. J., Beekmann, M., & Burrows, J. P. (**2010**). On the improvement of NO 2 satellite retrievals–aerosol impact on the airmass factors. Atmospheric Measurement Techniques, 3(2), 475-493.

**[References #5]**
**McLinden**, C. A., Fioletov, V., Boersma, K. F., Kharol, S. K., Krotkov, N., Lamsal, L., ... & Yang, K. (**2014**). Improved satellite retrievals of NO 2 and SO 2 over the Canadian oil sands and comparisons with surface measurements. Atmospheric Chemistry and Physics, 14(7), 3637-3656.

**[References #6]**
**Mei**, L. L., Xue, Y., Kokhanovsky, A. A., von Hoyningen-Huene, W., De Leeuw, G., & Burrows, J. P. (**2014**). Retrieval of aerosol optical depth over land surfaces from AVHRR data. Atmospheric Measurement Techniques, 7(8), 2411-2420.

**[References #7]**
**Chen**, L., Wang, R., Han, J., & Zha, Y. (**2021**). Influence of observation angle change on satellite retrieval of aerosol optical depth. Tellus B: Chemical and Physical Meteorology, 73(1), 1-14.

**[References #8]**
**Wang**, Y., Yuan, Q., Li, T., Shen, H., Zheng, L., & Zhang, L. (**2019**). Evaluation and comparison of MODIS Collection 6.1 aerosol optical depth against AERONET over regions in China with multifarious underlying surfaces. Atmospheric Environment, 200, 280-301.

**[References #9]**
**Platnick**, S., Li, J. Y., King, M. D., Gerber, H., & Hobbs, P. V. (**2001**). A solar reflectance method for retrieving the optical thickness and droplet size of liquid water clouds over snow and ice surfaces. Journal of Geophysical Research: Atmospheres, 106(D14), 15185-15199.

**[References #10]**
**Letu**, H., Yang, K., Nakajima, T. Y., Ishimoto, H., Nagao, T. M., Riedi, J., ... & Shi, J. (**2020**). High-resolution retrieval of cloud microphysical properties and surface solar radiation using Himawari-8/AHI next-generation geostationary satellite. Remote Sensing of Environment, 239, 111583.

> 5. P3 L60: References will be added with respective to the different products of LER.

➔ We have added the relevant references as suggested.The modified parts are as follows: (The previously written parts are in blue and italics, and the newly added/replaced parts are in red and italics.)

*(Line 70-72; In Section 1. Introduction)*
 *This is exemplified by the geometry-dependent surface Lambertian-equivalent reflectivity (GLER)* *(Vasilkov et al., 2017; Qin et al., 2019) and the directionally dependent Lambertian-equivalent reflectivity (DLER) (Tilstra et al., 2021, 2023).*

**[References #11]**
**Vasilkov**, A., Qin, W., Krotkov, N., Lamsal, L., Spurr, R., Haffner, D., ... & Marchenko, S. (**2017**). Accounting for the effects of surface BRDF on satellite cloud and trace-gas retrievals: a new approach based on geometry-dependent Lambertian equivalent reflectivity applied to OMI algorithms. Atmospheric Measurement Techniques, 10(1), 333-349.

**[References #12]**
**Qin**, W., Fasnacht, Z., Haffner, D., Vasilkov, A., Joiner, J., Krotkov, N., Fisher, B., and Spurr, R.: A geometry-dependent surface Lambertian-510 equivalent reflectivity product for UV–Vis retrievals–Part 1: Evaluation over land surfaces using measurements from OMI at 466 nm, Atmospheric Measurement Techniques, 12, 3997–4017, (**2019**).

**[References #13]**
**Tilstra**, L. G., Tuinder, O. N., Wang, P., and Stammes, P.: Directionally dependent Lambertian-equivalent reflectivity (DLER) of the Earth's surface measured by the GOME-2 satellite instruments, Atmospheric Measurement Techniques, 14, 4219–4238, (**2021**)

**[References #14]**
**Tilstra**, L. G., de Graaf, M., Trees, V., Litvinov, P., Dubovik, O., and Stammes, P.: A directional surface reflectance climatology determined from TROPOMI observations, Atmospheric Measurement Techniques Discussions, 2023, 1–29, (**2023**).

> 6. P3 L84-91: The author shows the suggestion of the methods. However, the purpose of this study and importance of study are not included in the Introduction. Please include the purpose of study and also describe the sections.

➔ We acknowledge that the previous explanation for this section was inadequate. In response to your feedback, we have now included the purpose and significance of this study, as well as a detailed explanation of each section at the conclusion of the introduction.

The revised text is provided below : (The previously written parts are in blue and italics, deleted parts are blue and , and the newly added/replaced parts are in red and italics.)

*(Line 99-107; In Section 1. Introduction)*
*Therefore, in this study, we propose, for the first time, the application of the BRDF model to hyperspectral satellite data for more realistic preliminary surface reflectance data. In this study, we focused on 440 nm, which is used as an input from $NO_2$, clouds, and aerosols. This algorithm consists of two main steps: atmospheric correction, BRDF modeling and BSR retrieval. The purpose of this study is to enhance the accuracy of satellite-derived prior surface reflectance data by addressing BRDF effects through a method that incorporates the strengths of both GLER and DLER while addressing their limitations. This is crucial for improving the reliability of atmospheric and surface property retrievals from hyperspectral satellite data. A detailed description of each step of the proposed algorithm is provided in Section 3. Section 2 covers the data and study area, and Section 4 presents the results and discussion. *

> 7. P4 Please change the order of section. Section 2.1 will be shown after the Section 2.2.

➔ We have revised the relevant sections as suggested. And also, we edited the title of Section 2. (Study area and Materials ➔ Materials and Study area). Thank you for your .

  Below, we have included the revised the order of sections for your review. The modified parts are as follows: (The previously written parts are in blue and italics, and the newly added/replaced parts are in red and italics.)

*(Line 108-149; In Section 2. Study area and Materials)*
*(Before the change)*
*2. Study area and Materials*
*2.1. Study area*
*2.2. Geostationary Environment Monitoring Spectrometer (GEMS)*

*(After the change)*
*2. Materials and Study area*
*2.1. Geostationary Environment Monitoring Spectrometer (GEMS)*
*2.2. Study area*

> 8. P4 L104: Level-1C ➔ Level 1C (L1C)

➔ We have revised the relevant sections as suggested. Thank you.

> 9. P4 L110: 'it does not provide an official cloud mask': The GEMS officially provides the cloud product (GEMS CLD). It is confusing to the reader. What is this sentence means?
>
> 10. P4 L110-111: For the clear-sky identification, is this study uses the CCP > 1000 hPa, addition to the ECF<0.2 and CCP = 1013hPa? It has advantage of clear-sky identification from falsely detected as cloudy pixels. However, how about the cloud conditions in real, but cloudy from GEMS CLD?

➔ We think it will be easier for you to understand the answers to the two questions regarding cloud products above if we explain them all at once.

Environmental satellites such as GEMS do not officially provide the cloud detection data that is provided by satellites with multispectral sensors such as MODIS and GK-2A. In this case, cloud detection data is a classification of whether or not clouds are present within a given pixel, often called a "cloud mask". Instead, we utilize a threshold in each output based on variables such as ECF (Effective Cloud Fraction) and CCP (Cloud Centroid Pressure) for classifying as clear-sky and cloud. Therefore, almost GEMS field algorithm use the ECF data to remove these cloud effect, and the threshold varies from 0.2 to 0.4.

The variable CCP indicates the location of clouds, and a CCP value closer to 1013 hPa means the clouds are located closer to the ground. For algorithms that calculate atmospheric aerosols and gases, the effect

of clouds close to the ground can be ignored, but for algorithms that calculate ground reflectance, undetected clouds can be a significant source of error.

Therefore, in the quality flag within GEMS CLD ATBD, when ECF is less than 0.2 or CCP is equal to 1013 hPa, it is classified as clear-sky. However, in this study, to remove clouds located very close to the ground, we have set the following criteria: (1) clear-sky if ECF is less than 0.2, and (2) cloud-sky if CCP is more than 1000 hPa, even if ECF is less than 0.2.

We recognize that this sentence may be confusing to other readers, and we have made the following additions and corrections to the text.

We have attached the revised text below. (The previously written parts are in blue and italics, deleted parts are blue and , and the newly added/replaced parts are in red and italics.)

*(Line 120-134; In Section 2. Geostationary Environment Monitoring Spectrometer (GEMS))*
*The GEMS CLD product is an optical quantity observed at UV-VIS wavelengths, which may differ from the physical properties of real clouds; therefore, it does not provide an official detection data like those from multi-spectral sensors such as MODIS (Frey et al., 2008) and GK-2A (Lee and Choi, 2021). Cloud detection data, often referred to as a "cloud mask," classifies whether clouds are present within a given pixel. Instead, GEMS utilizes thresholds based on variables such as Effective Cloud Fraction (ECF) and Cloud Centroid Pressure (CCP) to classify pixels as clear-sky or cloudy. Most GEMS field algorithms use ECF data to mitigate cloud effects, with thresholds varying from 0.2 to 0.4. Within the quality flag of GEMS CLD algorithm theoretical basis documents (ATBD) (Choi et al., 2020), pixels are classified as clear-sky when ECF is less than 0.2 or CCP is equal to 1013 hPa. CCP indicates the location of clouds, with values closer to 1013 hPa signifying clouds closer to the ground. While ground-level clouds can be ignored in algorithms calculating atmospheric aerosols and gases, they can significantly affect algorithms calculating ground reflectance. Therefore, in this study, we adopted the following criteria to exclude clouds very close to the ground: (1) pixels are classified as clear-sky if ECF is less than 0.2, and (2) pixels are classified as cloudy if CCP is greater than 1000 hPa, even if ECF is less than 0.2. ~~However, if the effective cloud fraction (ECF) is less than 0.2 and the cloud centroid pressure (CCP) is equal to 1013 hPa, the quality flag in the CLD output indicates a clear sky. However, to mitigate potential errors caused by lower clouds being mistaken as clear-sky, an additional cloud-masking step was performed when the CCP was equal to or greater than 1000 hPa, following the initial masking based on the ECF.~~*

**[References #15]**
*Frey, R. A., Ackerman, S. A., Liu, Y., Strabala, K. I., Zhang, H., Key, J. R., & Wang, X. (2008). Cloud detection with MODIS. Part I: Improvements in the MODIS cloud mask for collection 5. Journal of atmospheric and oceanic technology, 25(7), 1057-1072.*

**[References #16]**
*Lee, S., & Choi, J. (2021). Daytime cloud detection algorithm based on a multitemporal dataset for GK-2A imagery. Remote Sensing, 13(16), 3215.*

**[References #17]**
*Choi, Y.-S., Kim, G., Kim, B.-R., Kwon, M.-J., Kim, Y., Yoon, J., won Lee, D., and Kim, J.: Geostationary Environment Monitoring Spectrometer (GEMS) Algorithm Theoretical Basis Document:*

*Cloud Retrieval Algorithm, Tech. rep., Environmental Satellite Center, National Institute of Environmental Research, Ministry of Environment, South Korea, file:///mnt/data/atbd_cld.pdf, version 1.1, Approved by Jhoon Kim on 03 Feb 2021, )(**2020**).*

> 11. Change "R-value" to 'r'

➔ We have revised the relevant sections as suggested. Thank you.

> 12. P4 L116-117: The author shows the accuracy of GEMS AOD at 443 nm. However, I am doubtful that this accuracy results is not guaranteed different spectral AOD accuracy. During the spectral conversion of AOD, the error will be enhanced. Why don't the author use the 443 nm AOD values?

➔ As mentioned in the text, most satellite AOD algorithms calculate 550 nm AOD because the 550 nm wavelength is important because it is the most scattered in the atmosphere and is widely used in various chemical models. Therefore, the common reference wavelength for most satellite AOD products is set to 550 nm. The 6SV RTM used to perform the atmospheric correction in this study is also configured to input AOD values at 550 nm. Therefore, the AOD at 550 nm was used as an input in this study.

In the case of the GEMS AOD algorithm, the AOD at 443 nm is first calculated, and then the AOD at 550 nm is determined through wavelength conversion. This process may introduce errors, but no AOD errors due to wavelength conversion have been reported yet in the official GEMS AOD algorithm. However, due to the nature of the 6SV RTM, calculations can only be performed with the AOD at 550 nm as input, limiting the use of the 443 nm AOD.

I agree with your opinion and appreciate your valuable comment. It would be beneficial to conduct an analysis of surface reflectance considering the impact of wavelength conversion in future studies.

> 13. P5 L125-130: For the supplement of GEMS AOD gap, this study used the CAMS AOD values. However, the CAMS AOD have significant biases as compared to the GEMS AOD. The AOD bias between GEMS and CAMS will be affected to the discontinuous spatial distribution of AOD, and thus affecting to the surface reflectance calculation. How much this bias affecting the surface reflectance estimation? In addition, how did this study correct the AOD bias between CAMS and GEMS?

➔ We fully acknowledge that there are differences between CAMS AOD data and GEMS AOD data, and we agree with this observation. However, this study is a foundational effort to develop a working algorithm for calculating surface reflectivity (BSR) using GEMS satellite data. The additional use of CAMS AOD data was designed to enhance the accuracy of the reflectivity product and ensure the stability of the calculations.

When comparing CAMS AOD and GEMS AOD, we observed an RMSE difference of approximately 0.2, with GEMS tending to underestimate. When calculating TOC based on this data, the RMSE was about 0.015. Given that CAMS AOD data was used in a relatively small proportion of the pixels for BRDF modeling, we determined that this influence could be sufficiently mitigated. Despite the inherent differences between GEMS AOD and CAMS AOD data, we prioritized maintaining a complete dataset over potential biases.

Furthermore, qualitative analysis using CAMS AOD in the absence of GEMS AOD, prior to applying the algorithm, revealed no noticeable discontinuities, supporting the applicability of this method.

As the reviewer pointed out, we recognize that the AOD bias between the two datasets affects TOC, and this influence is subsequently transferred to BSR. However, it is important to emphasize that this study aims to develop a methodology for the current GEMS algorithm's surface reflectance calculation. While acknowledging the limitations of not addressing the bias between the GEMS and CAMS AOD datasets, this study provides a fundamental framework for operational use. Future research will focus on developing and integrating bias correction techniques to further improve calculation accuracy.

By transparently addressing these limitations and outlining plans for future improvements, we demonstrate our commitment to enhancing the reliability and precision of surface reflectance calculations from satellite data. The limitations of this aspect have been included in the final discussion section, and the last section, originally titled "Conclusion," has been revised to "Conclusion and Discussion."

We have attached the revised text below. (The previously written parts are in blue and italics, deleted parts are blue and , and the newly added/replaced parts are in red and italics.)

*(Line 466-482; In Section 5.Conclusion and Discussion )*
*(Before the change)*
*In conclusion, our study demonstrated that BSR can effectively simulate realistic reflectance, surpassing the minimum reflectance approach used in many existing studies. Although limitations exist, such as the challenge of capturing sudden changes in surface characteristics such as snow or ice cover, our research is pioneering in its application to BRDF modeling and evaluation in hypersprectral observation satellite studies. By combining the high temporal and spatial resolution of GLER with the BRDF considerations of DLER, we laid the foundation for improved accuracy in the AQ output. Our findings suggest that the utilization of BSR, a dataset reflecting realistic reflectance with BRDF effects, can enhance various climate analysis studies, marking a significant advancement in the field.*

*(After the change)*
*Although limitations exist, such as the challenge of capturing sudden changes in surface characteristics such as snow or ice cover. And also, To inhance stability, we designed a method to use CAMS AOD data in the absence of GEMS AOD, acknowledging the presence of some bias. Despite this limitation, we prioritized maintaining a stable dataset. Our research is pioneering in its application to BRDF modeling and evaluation in hyperspectral observation satellite studies. We are committed to further refining our approach and will strive to address these biases in future research to improve calculation accuracy.*

*In conclusion, our study demonstrated that BSR can effectively simulate realistic reflectance, surpassing the minimum reflectance approach used in many existing studies. By combining the high temporal and spatial resolution of GLER with the BRDF considerations of DLER, we laid the foundation for improved accuracy in the AQ output. Our findings suggest that the utilization of BSR, a dataset reflecting realistic reflectance with BRDF effects, can enhance various climate analysis studies, marking a significant advancement in the field.*

> 14. Section 3: Before beginning the section 3, the author needs to the definition and calculation process of Top-of-Canopy from GEMS.

➔ We recognized the lack of mention of this part and have added TOC-related sentences to the text.

We have attached the revised text below. (The previously written parts are in blue and italics, deleted parts are blue and , and the newly added/replaced parts are in red and italics.)

*(Line 170-174; In Section 3. Background Surface Reflectance (BSR) retrieval algorithm)*
*Figure 1 depicts a comprehensive flow chart of the BSR retrieval algorithm, which comprises two primary steps: (1) atmospheric correction and (2) BRDF modeling and BSR retrieval. The top-of-canopy (TOC) reflectance from GEMS represents the actual surface reflectance derived through atmospheric correction when AOD, CLD, and O3T products are available. To evaluate the applicability of the BSR derived in this study, validations were performed against the GEMS TOC*  *data as reference data.*

> 15. P5 L147 what is the 'traditional minimum reflectivity method'? Please add the details and references.

➔ The "minimum reflectance method," referred to in the text as the traditional minimum reflectivity method, is an algorithm that currently underpins various LER data. This method was first designed for the Total Ozone Mapping Spectrometer (TOMS) and was initially mentioned in the study by Eck et al. (1987). It involves selecting and using the minimum value among the surface reflectances observed over a certain period for each grid area. This approach effectively reduces the influence of temporary air pollution (such as aerosols and clouds) on the reflectivity measurements, thereby isolating the true surface reflectance.

$$R = \frac{A - A_0(\tau, \theta_0, \theta, \phi)}{f_1(\tau, \theta_0) f_2(\tau, \theta) + [A - A_0(\tau, \theta_0, \theta, \phi)] f_3(\tau)}$$

**equation 1. Lambertian Equivalent Surface Reflectance (LER) calculation formula**

It can be calculated using the above formula, and the minimum value among these calculated values is selected. Where A is the directional albedo, $A_0$ is the atmospheric albedo, and $f_1$, $f_2$, $f_3$ are fractions accounting for various scattering effects.

Therefore, references to this paper and the GLER and DLER papers that have recently been calculated based on this algorithm (**Eck et al., 1987**) have been added to the text. However, the word "traditional" was deleted from the text to avoid confusing readers with different algorithms.

Below is the edited text. (The previously written parts are in blue and italics, deleted parts are blue and , and the newly added/replaced parts are in red and italics.)

*(Line 174-176; In Section 3. Background Surface Reflectance (BSR) retrieval algorithm)*
*Additionally, a comparison was made with the LER data generated using the*  *minimum reflectivity method which is used for a variety of LER datasets (**Kleipool et al. (2008); Koelemeijer et al. (2003); Tilstra et al. (2017)**)) and was first introduced in the work of **Eck et al. (1987).***

**[References #18]**
**Kleipool**, Q. L., Dobber, M. R., de Haan, J., & Levelt, P. F. (**2008**). Earth surface reflectance climatology from 3 years of OMI data. Journal of Geophysical Research: Atmospheres, 113(D18).

**[References #19]**
**Koelemeijer**, R. B. A., De Haan, J. F., & Stammes, P. (**2003**). A database of spectral surface reflectivity in the range 335–772 nm derived from 5.5 years of GOME observations. Journal of Geophysical Research: Atmospheres, 108(D2).

**[References #20]**
**Tilstra**, L. G., Tuinder, O. N. E., Wang, P., & Stammes, P. (**2017**). Surface reflectivity climatologies from UV to NIR determined from Earth observations by GOME-2 and SCIAMACHY. Journal of Geophysical Research: Atmospheres, 122(7), 4084-4111.

**[References #21]**
**Eck**, T. F., Bhartia, P. K., Hwang, P. H., & Stowe, L. L. (**1987**). Reflectivity of Earth's surface and clouds in ultraviolet from satellite observations. Journal of Geophysical Research: Atmospheres, 92(D4), 4287-4296.
* * *
16. Figure 1: Change the caption to "Flowchart of GEMS BSR algorithm."
* * *
➔ We have revised the relevant sections as suggested. Thank you.
* * *
17. Equation 1-5 and related sentences: For the equation writing, please use the subscript. All the equation variables are not use the subscript and it may confusing to the equation. In addition, all the equation variables will be clarify in the manuscript.
* * *
➔ We have revised the relevant sections as suggested. Additionally, when variables written as formulas are mentioned in the text, they are written in italics, the same as the formula.

  Below, we have included the revised text and the updated figure for your review. The modified parts are as follows: (The previously written parts are in blue and italics, and the newly added/replaced parts are in red and italics.)

*(Line 189-193; In Section 3.1. Atmospheric correction)*
*  By employing the three atmospheric correction coefficients ($x_{ap}$, $x_b$, and $x_c$) derived from the 6SV RTM, the TOC reflectance can be calculated from the TOA reflectance. The surface reflectance was then calculated from the TOA reflectance, as expressed in Equation (1). The atmospheric correction coefficients $x_{ap}$, $x_b$, and $x_c$ can be computed using Equations (2), (3), and (4), where they represent the inverse of the transmittance, scattering term of the atmosphere, and spherical albedo, respectively.*

*(Line 223-226; In Section 3.2 BSR retrieval through BRDF modeling)*
*  The Roujean BRDF model defines surface reflectance as a combination of isotropic, geometric, and volumetric scattering components. It comprises two physical kernels ($f_1$ and $f_2$) and three empirical coefficients ($K_0$, $K_1$, $K_2$; BRDF parameters) that describe the mechanism of each component, as shown in Equation 5.*

[Figure]

**Figure 1. Schematic of 15-day composite period and retrieval cycle for BRDF modeling (Figure 2 in this article; Before the change)**

[Figure]

**Figure 2. Schematic of 15-day composite period and retrieval cycle for BRDF modeling (Figure 2 in this article; After the change)**

18. P6 L160: For the 6SV RTM, How to adopting the spectral response function of GEMS?

➔ *As mentioned in line 188 of the text (in article), the 6SV is calculated by dividing it into 2.5 nm intervals and summing the values afterwards. Therefore, for GEMS with a band width of 3.6 nm for one channel, we calculated it with a monochromatic wavelength value (center wavelength).*

*In addition, the LUTs for calculating the reflectance of the ground surface of other current satellites (TROPOMI and GOME-2 DLER) are mostly calculated at monochromatic wavelengths. Below are the papers of GOME-2 (**Tilstra et al., 2021**) and TROPOMI (**Tilstra et al., 2023**) DLER algorithms mentioned in this study, both of which are mentioned in table1.*

**\* Tilstra et al., 2021 : Same as [References #13 in this response letter]**
**\* Tilstra et al., 2021 : Same as [References #14 in this response letter]**

19-1 Table 1: For the Aerosol type, what is "continental"? Do you have any detailed aerosol optical and physical properties, or related references?

→ The "Continental" model utilized in this study is an aerosol model predefined within 6SV. The term "Continental" in the context of aerosol types refers to aerosol particles typically found in continental regions. This is a model produced by mixing three basic components: dust-like component (DUST), water-soluble component (WATE), and soot component (SOOT). The weighted averages of DUST, WATE, and SOOT in that order are 0.7 and 0.29, 0.01, respectively.

For the continental aerosol model, the manual specifies the following properties based on predefined components and their volume mixing ratios:

**(1) Dust-Like Component (D.L.):**
Volume Concentration: 113.98352 µm³/cm³
Particle Number Concentration: 54.734 part/cm³
Characteristics: Coarse mode particles with significant scattering properties.

**(2) Water-Soluble Component (W.S.):**
Volume Concentration: $113.98352 \times 10^{-6}$ µm³/cm³
Particle Number Concentration: $1.86850 \times 10^{6}$ part/cm³
Characteristics: Fine mode particles, often hygroscopic, affecting scattering and absorption.

**(3) Oceanic Component (O.C.):**
Volume Concentration: 5.14441 µm³/cm³
Particle Number Concentration: 276.0500010 part/cm³
Characteristics: Typically sea salt particles, contributing to scattering.

**(4) Soot Component (S.C.):**
Volume Concentration: $59.777553 \times 10^{-6}$ µm³/cm³
Particle Number Concentration: $1.86850 \times 10^{6}$ part/cm³
Characteristics: Strongly absorbing particles, influencing the single scattering albedo.

These properties and their computational parameters are discussed in the context of aerosol models in the 6S Manual, especially within the sections describing the aerosol optical properties, phase functions, and the mixing of different aerosol types based on the volume percentages(detailed in 6SV Manual Part2 (Vermote et al., 2006) ).

**[Reference #22]**
Vermote, E., Tanré, D., Deuzé, J. L., Herman, M., Morcrette, J. J., & Kotchenova, S. Y. (2006). Second Simulation of a Satellite Signal in the Solar Spectrum - Vector (6SV) User Guide (Version 3). University of Maryland, Laboratoire d'Optique Atmosphérique, European Centre for Medium Range Weather Forecast. Retrieved from 6S_Manual_Part_1, 6S_Manual_Part_2, 6S_Manual_Part_3.

> 19-2. In addition, the TCO value range is too narrow. In East Asia, the total ozone is ranged from 200-600 DU based on the daily data. 250-350 DU ranges are too narrow. Do this narrow TCO range affect the surface reflectance retrieval?

→ The range and interval settings of TCO when creating the LUT were heavily reliant on criteria from other satellites. Both COMS (**Lee et al., 2018**) and GK-2A (**Lee et al., 2020**), a geostationary satellite focused on observations over Asia like GEMS, land surface reflectance algorithm utilized the same TCO for LUT generation as our study.

Additionally, based on the findings of **Shin et al. (2021),** we confirmed that most of the ozone over East Asia is distributed within the range of 250-350 DU. This paper analyzed the trend of TCO from 1997 to 2017, with **Figure 1** in the paper illustrating the results.

While ozone levels can reach up to 600 DU, this study is a preliminary investigation for the application of the algorithm in the field. Therefore, the LUT was created with a focus on the efficiency of the calculations..

**[Reference #22]**
**Lee**, C. S., Han, K. S., Yeom, J. M., Lee, K. S., Seo, M., Hong, J., ... & Roujean, J. L. (**2018**). Surface albedo from the geostationary Communication, Ocean and Meteorological Satellite (COMS)/Meteorological Imager (MI) observation system. GIScience & Remote Sensing, 55(1), 38-62.

**[Reference #23]**
**Lee**, K. S., Chung, S. R., Lee, C., Seo, M., Choi, S., Seong, N. H., ... & Han, K. S. (**2020**). Development of land surface albedo algorithm for the GK-2A/AMI instrument. Remote Sensing, 12(15), 2500.

**[Reference #24]**
**Shin**, D., Oh, Y. S., Seo, W., Chung, C. Y., & Koo, J. H. (**2021**). Total ozone trends in east asia from long-term satellite and ground observations. Atmosphere, 12(8), 982.

> 20. P9 L231: This study uses the 15-day period for the clear-sky identification. However, 15-day is too narrow temporal window. From the below reference, the 30-day temporal window is essential to identify the clear-sky conditions. Park, S. S., Yu, J. E., Lim, H., and Lee, Y. G.: Temporal variation of surface reflectance and cloud fraction used to identify background aerosol retrieval information over East Asia, Atmos. Environ., 309, 119916.

→ The synthesis period for BRDF (Bidirectional Reflectance Distribution Function) modeling in this study was set to 15 days, as it was empirically determined to be optimal. The same period was applied for LER (Lambertian Equivalent Reflectance) calculation.

The referenced paper (**Park et al., 2023**) also employs the minimum reflectance method, which selects the minimum value within a grid over a given time window. They analyzed synthesis period settings of 15, 20, 30, and 40 days. The findings suggest that a 30-day temporal window is generally recommended for identifying clear-sky conditions due to cloud cover variability. However, the study also highlights the importance of considering surface reflectance variability when choosing the temporal window. It concludes that a 15-day window can effectively balance minimizing cloud contamination and maintaining consistent surface reflectance, as longer windows reduce cloud influence but increase surface reflectance variability.

Therefore, we determined that the 15-day compositing window set in this study is appropriate for both the accuracy of BRDF modeling and the mitigation of errors introduced by clouds when generating LERs.

Through the paper you mentioned, we were able to obtain a scientific basis for the 15-day synthesis period set in this study. We sincerely appreciate your guidance in this matter. Consequently, we have added the referenced paper to the main text and made the necessary modifications. Thank you once again for your valuable feedback.

The revised section is attached below. The modified parts are as follows: (The previously written parts are in blue and italics, and the newly added/replaced parts are in red and italics.)

*(Line 261-267; In Section 3.3. GEMS LER assumption)*
*(Before the change)*
*The "GEMS LER" was determined as the minimum reflectance over a 15-day period, aligning with the BRDF synthesis period. Additionally, the GEMS LER was used for further gap filling in cases of persistent missing in GEMS BSR, despite utilizing the age variable.*

*(After the change)*
*The "GEMS LER" was determined to be the minimum reflectance over a 15-day period to match the BRDF synthesis period. A 15-day synthesis period was also found by **Park et al. (2023)** to be an effective balance between minimizing cloud contamination and maintaining consistent surface reflectance in the application of the minimum reflectance method. The resulting GEMS LER was used for additional gap filling in cases where the GEMS BSR was consistently missing despite utilizing the age variable.*

**[Reference #25]**
**Park**, S. S., Yu, J. E., Lim, H., & Lee, Y. G. (**2023**). Temporal variation of surface reflectance and cloud fraction used to identify background aerosol retrieval information over East Asia. Atmospheric Environment, 309, 119916.

> 21. Figure 3: The intercomparison has been done by using the GEMS TOC. By the comparison with GEMS TOC, how to be explain the significance of accuracy improvements?

➔ As mentioned in the introduction, using LER based on the minimum reflectance method as an alternative reflectance causes an underestimation of surface reflectance, leading directly to the overestimation of AOD and affecting other L2 products such as clouds and gas products.

Both BSR and LER are meaningful variables that have already simulated values almost similar to TOC reflectance, in advance.

As highlighted in section 4.2.2, "Surface Reflectance Influence on AOD Variability in Cropland and Urban Areas," while the overall accuracy improves by approximately 3%, the impact on AOD shows that BSR is up to about 10% more accurate than LER, especially in urban areas during winter.

Therefore, the enhancement of BSR is significant as it addresses the inherent issues associated with LER.

> 22. P11 L275: GEMS SFC is not defined in this manuscript.

➔ SFC stands for Surface Reflectance, which in this study is synonymous with TOC (Top-Of-Canopy) Reflectance. However, for consistency throughout the paper, we have changed it to TOC Reflectance. Thank you.

> 23. P13 L299: Please add the references and detailed products used in this study.

➔ The analysis was conducted based on MODIS land cover data. The data used is from the MODIS/Terra+Aqua Land Cover Type Yearly L3 Global 0.05Deg CMG V061 (MCD12C1) (**Strahler et al.,1994**), specifically from the year 2021. We used the land cover according to the IGBP classification within the MODIS Land Cover data. The values for the four land cover types used in this study are Grassland, Cropland, Shrubland and Urban. The average values for all four land cover types were utilized.

 We acknowledged that this explanation was insufficient and additionally mentioned the MODIS land cover datasets as follows. (The previously written parts are in blue and italics, and the newly added/replaced parts are in red and italics.)

*(Line 333-336; In Section 4.2.1. Time series consistency analysis by land types)*
*To assess the simulation performance based on the time series of the BSR, we analyzed the time series stability across four land types (grassland, cropland, shrubland, and urban) using MODIS land cover data. The dataset utilized in this study is derived from the MODIS/Terra+Aqua Land Cover Type Yearly L3 Global 0.05Deg CMG V061 (MCD12C1) product (**Strahler et al.,1994**). The classification follows the International Geosphere-Biosphere Programme (IGBP) scheme within the MODIS Land cover dataset. Specifically, we employed the land cover data from the year 2021*

**[Reference #26]**
**Strahler**, A., Moody, A., Lambin, E., Huete, A., Justice, C., Muller, J., Running, S., Salomonson, V., Vanderbilt, V., and Wan, Z.: MODIS land cover product: Algorithm theoretical basis document, MODIS documentation, (**1994**)

> 24. P15 L315: From Table 1, the AOD bin range is not exceeded to 1.5. How to be analyzed the AOD ranges up to 2.0?

➔ AOD values above 1.5 were changed to 1.5 and calculated, but were marked as bad quality with their own quality flag and not used as input for analysis and BRDF modeling. AOD above 1.5 is not only rare, but other satellites such as VIIRS and GOES-R currently define 0.8 and above as a high AOD condition. In addition, according to the validation report of the SEVIRI AOD algorithm (CM SAF, 2017), the SEVIRI satellite also specified a maximum value of 1.5 for AOD in its reflectance calculation (As shown in the Figure 3).

**[Reference #27]**
Clerbaux, N.; Ipe, A.; De Bock, V.; Urbain, M.; Baudrez, E.; Velazquez-Blazquez, A.; Akkermans, T.; Moreels, J.; Hollmann, R.; Selbach, N. **CM SAF** Aerosol Optical Depth (AOD) Data Record–Edition 1. Satell. Appl. Facil. Clim. Monit. **2017,** doi:10.5676/EUM_SAF_CM/MSG_AOD/V001.

**Table 2 : Values used in the LDA LUTs**

| | Parameters and unit | Values used for RTM simulations |
|---|---|---|
| AOD | Aerosol Optical Depth at 550nm (unit less) | 0.05  0.1  0.15  0.2  0.3  0.4  0.5  0.6  0.7  0.8  0.9  1.0  1.25  1.5 |
| k | RPV parameter controlling the bowl/bell shape (Minnaert function) (unit less) | 0.3 to 1.0 by step of 0.05 |
| theta | RPV parameter controlling the forward/backward scattering (unit | -0.35 to 0.05 by step of 0.05 |

**Figure 3. AOD values used in SEVIRI AOD algorithm (CM SAF, validation report (2017))**

25. Figure 8: Is it possible to include the direct comparison between GEMS BSR and LERs?

➜ Figure 8 (in this article) presents a comparison between TOC (the reference data in this study) and GEMS BSR, OMI GLER, and TROPOMI DLER data. And also, Figure 9 (in this article) shows a comparison of BSR with GLER and BSR with DLER.

These figures are attached below (Fig1 and Fig 2 in this letter).

However, to prevent any confusion regarding the order, a brief explanation of the figures has been added before discussing Figures 8 and 9 in the text.

The edited parts are as follows: (The previously written parts are in blue and italics, and the newly added/replaced parts are in red and italics.)

*(Line 411-412; In Section 4.4. Intercomparison between BSR and LER database)*
*Figure 8 compares GEMS TOC with GEMS BSR, TROPOMI DLER, and OMI GLER. Figure 9 presents the intercomparison between BSR and OMI GLER, as well as BSR and TROPOMI DLER.*

[Figure]

**Fig 1. Figure 8 in this article (Distribution density plot of GEMS BSR, TROPOMI DLER, and OMI**

**GLER based on GEMS TOC observations. (a) GEMS BSR; (b) TROPOMI DLER; (c) OMI GLER)**

[Figure]

**Fig 2. Figure 9 in this article (Quantitative comparison of GEMS BSR, TROPOMI DLER, and OMI GLER value distributions (a) Density plot of BSR and GLER; (b) Density plot of BSR and DLER; (c) Histogram of BSR and GLER; (d) Histogram of BSR and DLER)**